# Let-7 enhances murine anti-tumor CD8 T cell responses by promoting memory and antagonizing terminal differentiation

Alexandria C. Wells[1,4], Kaito A. Hioki [1,2,4], Constance C. Angelou[1,4], Adam C. Lynch[1], Xueting Liang [1], Daniel J. Ryan[1], Iris Thesmar[1], Saule Zhanybekova[3], Saulius Zuklys[3], Jacob Ullom[1], Agnes Cheong[1], Jesse Mager[1], Georg A. Hollander [3], Elena L. Pobezinskaya [1] ✉ & Leonid A. Pobezinsky [1] ✉

The success of the CD8 T cell-mediated immune response against infections and tumors depends on the formation of a long-lived memory pool, and the protection of effector cells from exhaustion. The advent of checkpoint blockade therapy has significantly improved anti-tumor therapeutic outcomes by reversing CD8 T cell exhaustion, but fails to generate effector cells with memory potential. Here, using in vivo mouse models, we show that let-7 miRNAs determine CD8 T cell fate, where maintenance of let-7 expression during early cell activation results in memory CD8 T cell formation and tumor clearance. Conversely, let-7-deficiency promotes the generation of a terminal effector population that becomes vulnerable to exhaustion and cell death in immunosuppressive environments and fails to reject tumors. Mechanistically, let-7 restrains metabolic changes that occur during T cell activation through the inhibition of the PI3K/AKT/mTOR signaling pathway and production of reactive oxygen species, potent drivers of terminal differentiation and exhaustion. Thus, our results reveal a role for let-7 in the time-sensitive support of memory formation and the protection of effector cells from exhaustion. Overall, our data suggest a strategy in developing next-generation immunotherapies by preserving the multipotency of effector cells rather than enhancing the efficacy of differentiation.

After antigen stimulation, naive CD8 T cells rapidly differentiate into effector cells, cytotoxic T lymphocytes (CTLs), the majority of which die upon antigen clearance, while surviving cells differentiate into highly protective, long-lived memory cells[1]. In addition to effector cytokines, CTLs express and store cytotoxic molecules such as perforin, granulysin (in humans) and granzymes in lytic granules, the secretion of which is triggered by the recognition of target host cells, and leads to the induction of their programed cell death[2–5]. In certain

pathological conditions, such as chronic infection and cancer, CTLs acquire "an exhausted state", marked by the gradual loss of effector function, cytokine expression and cytotoxic potency[6–8]. With the exception of a subpopulation of precursors, exhausted CTLs are terminally differentiated cells and have definitive epigenetic and transcriptional signatures. Current immunotherapies for cancer treatment are focused on strengthening the cytotoxic function of CTLs by preventing or reversing the exhausted state of effector cells via

[1]Department of Veterinary and Animal science, University of Massachusetts, Amherst, MA, USA. [2]UMass Biotech Training Program (BTP), Amherst, MA, USA. [3]Pediatric Immunology, Department of Biomedicine, University of Basel and University Children's Hospital Basel, Basel, Switzerland. [4]These authors contributed equally: Alexandria C. Wells, Kaito A. Hioki, Constance C. Angelou. ✉e-mail: pobezinskaya@umass.edu; lpobezinsky@umass.edu

blockade of immunosuppressive ligand-receptor interactions in the tumor microenvironment. Specifically, it has been shown that the multipotent precursors of terminal effector cells are the most responsive to this treatment and can differentiate into effector CTLs, while terminally differentiated exhausted cells are not rescued and fail to mount productive responses[9–11]. Furthermore, recent reports suggest that the less differentiated precursors of CTLs, which still retain memory potential, elicit the most efficient anti-tumor immune responses and are critical for the success of immunotherapies[9,12–16]. Therefore, the molecular mechanisms that regulate these processes are the subject of intensive research[17–20].

The let-7 miRNAs are an evolutionarily conserved large family of non-coding RNAs that are expressed in naive T cells[21–23]. Upon antigen stimulation, T cells quickly downregulate let-7 expression, which promotes proliferation and differentiation of activated T cells into effector lymphocyte subsets[24,25]. In CD8 T cells, lin28-mediated let-7 deficiency leads to enhanced cytotoxic activity of lin28Tg CTLs in vitro, whereas let-7 overexpression restricts this activity[25]. Here, we demonstrate the impact of let-7 miRNAs on the fate of CD8 T cells in vivo, where let-7 promotes formation of memory cells, while antagonizing terminal differentiation of effector CD8 T cells. Mechanistically, we found that let-7 inhibits mTOR activation, which is critical for production of reactive oxygen species (ROS) to drive terminal differentiation of activated CD8 T cells. This work also provides insight into the temporal requirements of this molecular mechanism, and may ultimately inform approaches for enhancing current immunotherapeutic strategies.

## Results

### Let-7 supports anti-tumor CTL activity and drives changes in the transcriptome

Let-7-deficient (lin28Tg) CTLs exhibited superior cytotoxic function in vitro[25]. We therefore hypothesized that lin28Tg CTLs should augment anti-tumor immunity in vivo[26]. To test the function of lin28Tg CTLs in anti-tumor responses, B16F10 mouse melanoma cells were genetically engineered to co-express a GFP reporter and the gp33-41 peptide (Supplementary Fig. 1a, referred to as B16$^{gp33}$) from the lymphocytic choriomeningitis virus (LCMV), such that tumor-specific lysis can be assessed using P14 T cell receptor transgenic CD8 T cells which recognize the gp33-41 peptide in the context of the D$^{b}$ molecule[27]. P14 CTLs generated from wild type (WT), lin28Tg, or let-7Tg mice lysed B16$^{gp33}$ cells in vitro with different efficiencies, as previously reported[25] (Supplementary Fig. 1b), confirming the inhibitory effect of let-7 miRNAs on the cytotoxic function of CTLs. However, upon adoptive transfer into B16$^{gp33}$ tumor-bearing mice, the same in vitro-generated lin28Tg CTLs failed to control tumor growth. Rather, let-7Tg CTLs provided the most robust tumor protection in vivo, where mice which received let-7Tg CTLs were the sole survivors of the study (Fig. 1a, b and Supplementary Fig. 1c). Importantly, tumors that did grow in mice injected with let-7Tg CTLs were GFP-, indicating that outgrowth was due to a loss of gp33 expression (Supplementary Fig. 1d). These results demonstrate that overexpression of let-7, rather than let-7-deficiency, enhances tumor rejection, despite opposing efficacy in vitro.

To address the emergent paradox between the in vitro and in vivo performance of lin28Tg and let-7Tg CTLs, the transcriptomes of in vitro-generated CTLs were analyzed. Principle component analysis highlighted distinct differences in the transcriptional profiles of P14 CTL populations from lin28Tg, let-7Tg and WT mice, where principal component 1 (PC1) and PC2 accounted for most of the total variance, 74% and 13%, respectively (Fig. 1c). All the populations were significantly separated, with let-7Tg and lin28Tg cells distinctly clustered from the WT population, suggesting that indeed, let-7 expression alters the CTL transcriptome. By comparing P14 let-7Tg and lin28Tg CTLs with P14 WT CTLs, we identified 1378 differentially expressed

genes (DEGs), where 216 genes were downregulated and 286 genes were upregulated in let-7Tg CTLs, while in lin28Tg CTLs 448 genes were downregulated and 428 were upregulated (Fig. 1d). The most differentially expressed genes in these data sets were lin28 and Col1A1 (a gene in the locus of which let-7Tg is inserted)[28], in lin28Tg and let-7Tg CTLs, respectively, confirming sample identities. We identified and clustered 177 DEGs, which were significantly differentially expressed across the sample groups (Fig. 1e, Supplementary data 1). To identify the transcriptional programs that are regulated by let-7 miRNAs in CTLs, we focused on two main clusters: cluster-1 contained the genes that were suppressed by let-7, while cluster-2 comprised the genes that were induced in the presence of let-7 (Fig. 1e). We noticed that these clusters contained key genes related to distinct lineages of differentiated CD8 T cells. In fact, gene set enrichment analysis (GSEA) revealed that lin28Tg CTLs expressed an effector gene signature that was entirely suppressed in let-7Tg CTLs (Fig. 1f-left panel and Supplementary Fig. 2a). Conversely, let-7Tg CTLs had a very pronounced memory signature that was absent in lin28Tg CTLs (Fig. 1f-right panel and Supplementary Fig. 2b). Specifically, lin28Tg CTLs expressed genes consistent with the terminally differentiated state (Fig. 1g) including transcription factors (Eomes, Id2, Prdm1, Myb, Batf, Irf8 and Ikzf2), effector molecules (Infg, Prf1, Gzma, Gzmb, Gzmk, Gzmm, Tnf, Fasl, Il10 and Wnt10b), inhibitory receptors (Entpd1, Havcr2, Pdcd1, Cd160 and Cd244a, in red) and other well-known effector markers (CX3CR1, Klrg1, Tnfrsf9)[10,29–40]. On the contrary, let-7Tg CTLs expressed genes such as memory-specifying transcription factors (Tcf7, Lef1, Foxo1, Id3, Bach2 and Klf2), cytokine and homing receptors (Il7ra, Sell, Cxcr3, Ccr7 and S1pr1), costimulatory receptors (Cd27 and Cd28) and the survival factor Bcl-2[32,41–54] (Fig. 1h). Furthermore, let-7Tg CTLs robustly upregulated genes associated with stem cell memory T cells (Fig. 1h, in red), a subpopulation of memory T cells characterized by enhanced homeostatic persistence, as well as a multipotent differentiation potential that facilitates the rapid generation of several protective memory and effector T cell populations[55,56]. Some of the key genes from both clusters were further validated on RNA and protein levels (Fig. 1i, j and Supplementary Fig. 3a, b, c), where only a few tested markers (CD27, CD28 and CXCR3 receptors) show no significant difference in surface expression. Of note, the expression of the costimulatory receptors 4-1BB (Tnfrsf9)[57] and OX40R (Tnfrsf4) was also sensitive to the modulation of let-7 concentrations in CTLs (Fig. 1h and Supplementary Fig. 3a). Finally, GO (gene ontology) enrichment analysis revealed that the transcriptome of lin28Tg CTLs is skewed towards generating robust effector responses, while that of let-7Tg CTLs prioritizes survival (Supplementary Fig. 3d).

The metabolic switch to aerobic glycolysis, known as the Warburg effect, is typical for cancer cells and also supports the growth of differentiating effector CD8 T cells, but is reduced in cells committed to the memory lineage[58–61]. Therefore, we measured the let-7-mediated impact on extracellular acidification rate (ECAR) in CTLs as a proxy of glycolysis. Let-7Tg CTLs exhibited low levels of ECAR in contrast to lin28Tg CTLs, where ECAR was increased in comparison to control wild-type CTLs (Supplementary Fig. 3e), suggesting that let-7 further supports differentiation of the memory lineage by restricting glycolysis[62–64]. We also noted that based on high oxygen consumption rate (OCR) and increased mitochondrial membrane potential (measured by JC-1 and TMRE incorporation), lin28Tg CTLs had elevated levels of oxidative phosphorylation and energized mitochondria, indicative of the overall high metabolic state of these cells (Supplementary Fig. 3e–h). Importantly, based on the markers, the observed phenotype of terminally differentiated lin28Tg CTLs results from let-7-deficiency, as it was reversed in lin28Tg CTLs forced to transgenically express let-7 (designated as lin28Tglet-7Tg) (Supplementary Fig. 4a, b). Ultimately, these data demonstrate that the expression of let-7 miRNAs promotes the transcriptional program for memory T cells while restraining the terminal effector program.

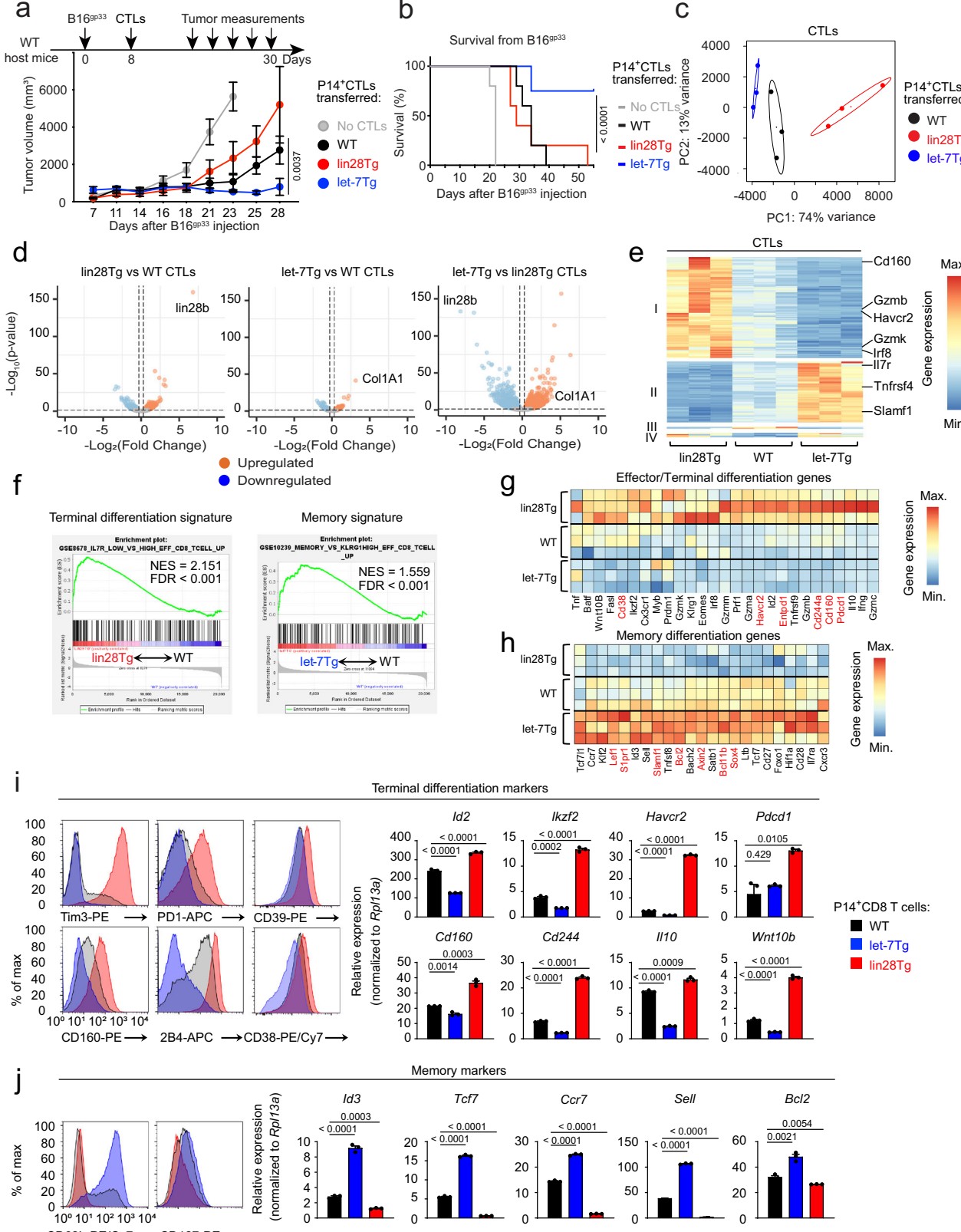

## Let-7 deficiency results in CD8 T lymphocyte dysfunction and cell death

Lin28Tg CTLs overexpressed many genes associated with exhaustion, including inhibitory receptors (Fig. 1g, in red), the engagement of which in the tumor microenvironment (TME) may explain the failure of these cells to control tumor growth in vivo (Fig. 1a, b). To determine

whether let-7 regulates the sensitivity of CD8 T cells to exhaustion, we examined the phenotypes of donor P14 CTLs with various levels of let-7 expression after injection into B16gp33 tumor-bearing mice. Recovered lin28Tg CTLs expressed high levels of the inhibitory receptors Tim3 and PD1, and were practically absent in the population of tumor infiltrating lymphocytes (TILs), as well as in the periphery of the recipient

**Fig. 1 | Let-7 promotes CTL anti-tumor response and changes the expression of key genes involved in CD8 T cell differentiation. a, b** Experimental design (**a**, top), tumor growth curves (**a**, bottom) and Kaplan–Meier survival curves (**b**) of WT mice s.c. injected with B16[gp33] tumor cells and adoptively transferred with P14[+] CTLs from either WT (*n* = 5), lin28Tg (*n* = 5) or let-7Tg (*n* = 4) mice. Gray color represents control group that received no CTLs (*n* = 5). **c–h**, RNA-seq of day 5 P14[+] CTLs from either WT, lin28Tg or let-7Tg mice. Data are from one RNA-seq analysis comprised of three biological replicates per group. **c** Principal component (PC) analysis of all differentially expressed genes (DEGs). **d** Volcano plots of all upregulated and downregulated DEGs in lin28Tg (left) and let-7Tg (middle) CTLs in comparison to WT CTLs and let-7Tg in comparison to lin28Tg CTLs (right). **e** Cluster analysis of top 177 upregulated and downregulated DEGs. **f** Enrichment for terminal differentiation signature genes in lin28Tg P14[+] CTLs and memory signature genes in let-7Tg P14[+] CTLs. FDR false discovery rate, NES normalized enrichment score. **g, h** Heatmap of terminal differentiation genes (**g**) and memory differentiation genes (**h**) in P14[+] CTLs. Red color represents the genes for inhibitory receptors (**g**) and stem cell signature genes (**h**). **i, j** FACS of the expression of proteins (left) and quantitative RT-PCR analysis of the expression of genes (right) involved in terminal differentiation (**i**) or memory formation (**j**) in WT, let-7Tg, and Lin28Tg CTLs. Data **i, j** are the means ± s.e.m. of technical triplicates; *P*-values were determined using a two-tailed unpaired Student's *t*-test (**i, j**), two-way ANOVA with Sidak's multiple comparison test (**a**) and log-rank Mantel-Cox test (**b**). Data represent two (**a, b**) or three (**i, j**) independent experiments. Source data for **a, b, i, j** are provided as a Source Data file.

mice (Fig. 2a, b). Conversely, let-7Tg CTLs had low expression of examined inhibitory receptors and comprised a large proportion of TILs. This phenotype was not tumor-specific as similar results were obtained using the EL4[gp33] tumor model (Supplementary Fig. 5a, b, c). The rapid loss of lin28Tg CTLs after adoptive transfer could result from a survival defect (Supplementary Fig. 3d) that we have previously reported for let-7 deficient naive T cells[65]. Therefore, prior to the transfer into tumor-bearing mice, lin28Tg CTLs were transduced with *bcl2l1*, the gene that encodes the pro-survival factor Bcl-xL (Fig. 2c). The expression of Bcl-xL significantly improved the number of lin28Tg CTLs in TILs but failed to enhance their anti-tumor performance (Fig. 2c and Supplementary Fig. 5d). Finally, we measured production of effector cytokines such as TNFα and IFNγ in recovered donor cells (Fig. 2d and Supplementary Fig. 5e). It appeared that only let-7Tg CTLs retained a significant ability to produce both cytokines, suggesting that let-7 supports not only survival, but also the polyfunctional state of effector CD8 T cells in the TME.

To assess the contribution of the immunosuppressive TME to the inactivation of lin28Tg CTL responses, anti-PDL1 antibodies were administered to B16[gp33] tumor-bearing mice after transfer of P14 lin28Tg and let-7Tg CTLs. Anti-PDL1 treatment significantly rescued lin28Tg CTL function, enhancing the anti-tumor response in a manner comparable to let-7Tg CTLs (Fig. 2e), indicating that the observed dysfunction of lin28Tg CTLs in vivo is in fact due to exhaustion induced by the immunosuppressive TME. Of note, the performance of let-7Tg CTLs was even further enhanced with anti-PDL1 treatment (Fig. 2e). Similar results were obtained using EL4[gp33] tumor model (Supplementary Fig. 5f).

To validate our findings on the polyclonal population of CD8 T cells and to eliminate any potential alterations in the T cell repertoire due to early expression of lin28 transgene in the thymus[66], we generated a new mouse model GzmbCre[+]R26[STOP-Lin28-GFP] of inducible let-7 knockdown only in activated CD8 T cells, such that lin28 expression is induced in Cre-positive cells and reported by GFP (Supplementary Fig. 6a, b). These mice were then inoculated subcutaneously with the MC57 fibrosarcoma, which is successfully controlled in syngeneic C57BL/6 mice. Tumor rejection was delayed and partially compromised by depleting let-7 miRNAs in responding CD8 T cells (Fig. 2f), confirming the critical role of let-7 deficiency in the dysfunction of CTLs upon activation in vivo. Next, to determine if there is a specific point during which loss of let-7 expression redirects differentiating CTLs to a terminal effector fate, R26[STOP-Lin28-GFP] mice were crossed to mice with a doxycycline inducible Cre and P14 TCR, generating P14Tg iCre[+]R26[STOP-Lin28-GFP] mice (Supplementary Fig. 6c), where lin28 expression can be permanently induced at any time during CTL differentiation. Surprisingly, regardless of when let-7 expression was depleted via lin28 induction, markers of terminal differentiation were upregulated, while memory phenotypic markers were downregulated (Fig. 2g). Altogether, these results indicate that the terminal effector fate is established upon let-7 deletion, making differentiated cells prone to exhaustion and even to cell death.

## Let-7 expression reprograms CD8 T cells into memory cells and promotes survival

Based on our results we hypothesized that let-7 may play a role in the differentiation of memory CD8 T cells. An important feature of memory cells is the ability to survive contraction after antigen clearance. Consistent with the upregulation of a pro-survival gene network (Fig. 1 and Supplementary Fig. 3c), expression of the let-7 transgene improved CTL viability during cytokine withdrawal in vitro[12] (Fig. 3a). Following contraction, cytokines such as IL−15 and IL-7 promote memory T cell differentiation and survival, and can be used to generate memory CD8 T cells in vitro[45,49,67,68]. As expected, WT CTLs upregulated the expression of *Tcf7*, *Sell*, *Ccr7*, and *CD127* only in the presence of IL-15, whereas let-7Tg CTLs displayed an enhanced memory phenotype regardless of the use of IL-2 or IL-15 (Fig. 3b). Although lin28Tg CTLs did not robustly upregulate expression of *Id2*, *Havcr2*, *Pdcd1*, and *Cd244* in the presence of IL-15, the induction of memory markers also failed (Fig. 3b). Similar results were also obtained with IL-7 cytokine (Supplementary Fig. 7a). Taken together, these in vitro results suggest that let-7 miRNAs are an essential component of both survival during contraction and memory differentiation, and that cytokines such as IL-15 are not sufficient to compensate for let-7 deficiency. To assess the impact of let-7 miRNAs on the generation of CD8 T cell memory in vivo, naive P14 CD8 T cells from WT, let-7Tg or lin28Tg mice were adoptively transferred into *Rag2-/-* hosts, which were subsequently infected with *Listeria monocytogenes* expressing the LCMV peptide gp33 (*Lm*-gp33)[69,70]. On days nine and twelve post-infection, the majority of donor P14 let-7Tg CD8 T cells displayed a phenotype of memory precursor effector cells (MPECs: KLRG1[-]CD127[+]), while a high proportion of P14 lin28Tg lymphocytes were short lived effector cells (SLECs: KLRG1[+]CD127[-]). This observation was further supported by the distribution of CD44 and CD62L expression[45] (Fig. 3c, Supplementary Fig. 7b). Moreover, by day thirty, a significant proportion of memory donor lin28Tg CD8 T cells retained an effector phenotype (Fig. 3c). Conversely, let-7Tg CD8 T cells exhibited the phenotype of central memory cells (CD44[+]CD62L[+]), demonstrating that let-7 promotes the generation of a memory CD8 T cell population in vivo. We also noticed that let-7 expression supported the survival of memory T cells generated in vivo, although the changes did not reach statistical significance (Supplementary Fig. 7c).

It has been suggested that cues essential for the formation of memory cells occur throughout CD8 T cell differentiation[1,71–75]. To determine if the timing of let-7 expression is important for its enhancement of memory formation, we took advantage of the fact that the expression of let-7Tg is doxycycline inducible and can be upregulated at different time points during CTL differentiation (Fig. 3d). As we have shown in Fig. 1, maintaining let-7 expression for all five days of CTL differentiation resulted in high levels of *Tcf7*, *Sell*, and *Ccr7* expression, and downregulation of *Id2*, *Cd244*, and *Havcr2* (Fig. 3d). However, limiting let-7 overexpression to the first 48 h of stimulation, was sufficient to upregulate expression of memory markers, and downregulate genes that contribute to terminal differentiation (Fig. 3d). Interestingly, inducing let-7

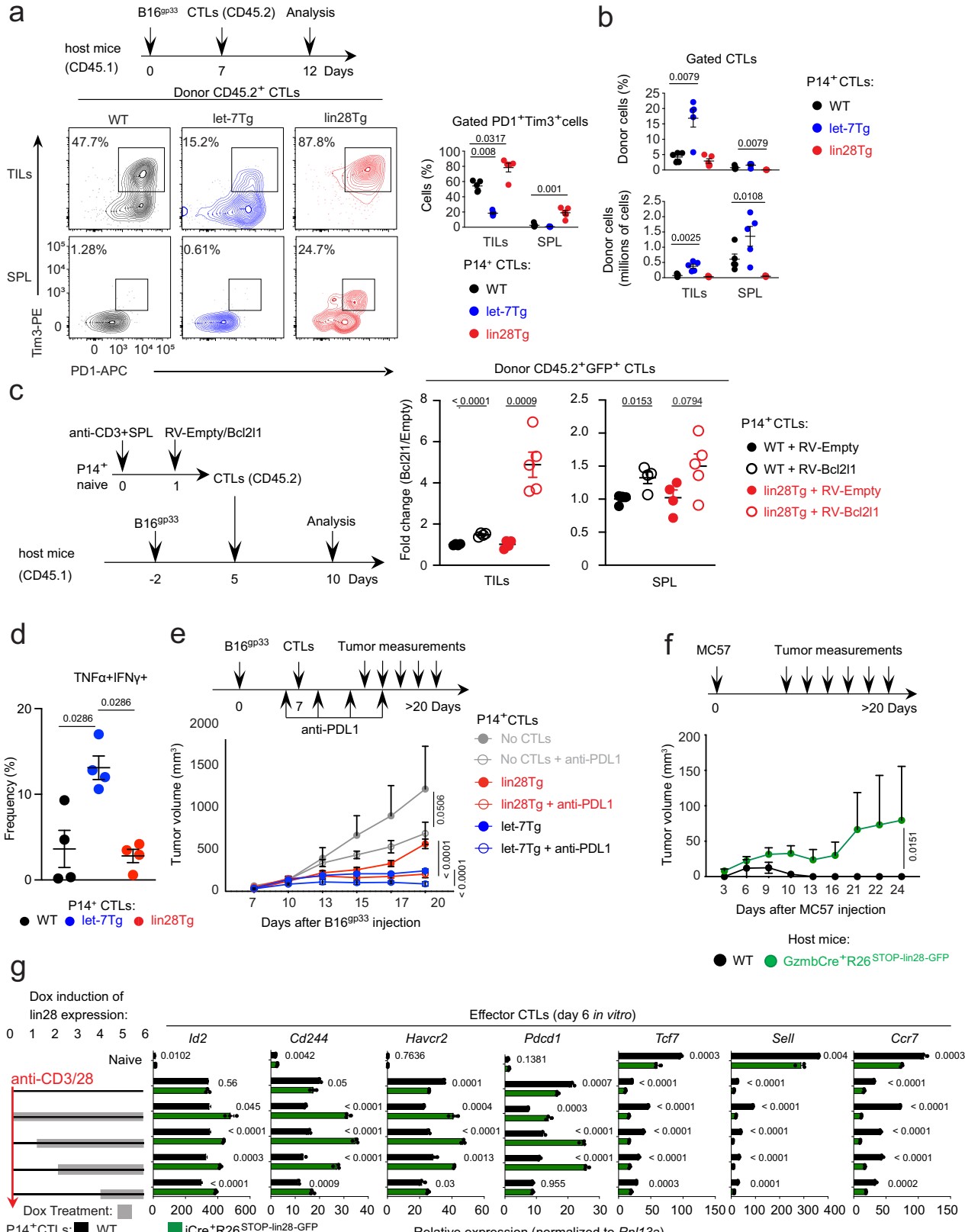

expression after the first 48 h of TCR stimulation had a much smaller effect (Supplementary Fig. 7d). Taken together, these results demonstrate that let-7 expression, specifically within the first hours of activation, is necessary for the generation of memory CD8 T cells, supporting studies illustrating that the memory fate is established soon after antigen stimulation[2,13,76–80].

## Let-7 regulates CD8 T cell fate through suppression of mTOR/ROS axis

Our results emphasize the role of let-7 miRNAs in CD8 T cell fate decisions during early antigen activation. To uncover the underlying molecular mechanisms of how let-7 promotes formation of immunological memory and restrains terminal differentiation, we compared

**Fig. 2 | Let-7-deficient CTLs undergo exhaustion in the tumor microenvironment. a, b** Experimental design and representative FACS analysis of surface expression of PD1 and Tim3 (**a**, left), quantification of the frequency of PD1⁺Tim3⁺ populations (**a**, right) and frequency and absolute number of donor CD45.2 P14⁺ CTLs (**b**) in TILs and spleens (SPL) of WT CD45.1 mice (*n* = 5 per group) s.c. injected with B16^gp33 tumor cells and adoptively transferred with donor WT, let-7Tg or lin28Tg P14⁺ CTLs. **c** Experimental design (left) and quantification of donor CD45.2 P14⁺ CTLs isolated from TILs or spleens of WT CD45.1 mice s.c. injected with B16^gp33 tumor cells and adoptively transferred with WT or lin28Tg P14⁺ CTLs transduced with either empty or Bcl2l1-encoding retroviruses (RV) (*n* = 4 per group, lin28Tg Bcl2l1-expressing P14⁺ CTL group was *n* = 5). **d** Quantification of the frequency of TNFα⁺IFNγ⁺ population in TILs of WT CD45.1 mice s.c. injected with B16^gp33 tumor cells and adoptively transferred with donor WT, let-7Tg or lin28Tg P14⁺ CTLs. TILs were restimulated for 4 h with PMA and ionomycin before intracellular cytokine staining. **e** Experimental design (top) and tumor growth curves (bottom) in WT

mice s.c. injected with B16^gp33 tumor cells and adoptively transferred with lin28Tg P14⁺ CTLs with (*n* = 8 mice) or without (*n* = 8 mice) anti-PDL1 treatment and let-7Tg P14⁺ CTLs with (*n* = 6 mice) or without (*n* = 9 mice) anti-PDL1 treatment. Gray color represents control group that received no CTLs (*n* = 6 mice). **f** Experimental design (top) and tumor growth curves (bottom) in GzmbCre⁻R26⁻ and GzmbCre⁺R26^Stop-Lin28-GFP mice s.c injected with MC57 tumor cells (*n* = 4 per group). Age and sex matched littermates were used. **g** Quantitative RT-PCR analysis of the expression of genes involved in terminal differentiation or memory formation in P14⁺ CTLs generated from control and iCre⁺R26^Stop-Lin28-GFP mice with addition of dox at indicated time points (gray bars). Data are the means ± s.e.m. of technical triplicates; *P*-values were determined using a two-tailed unpaired Student's *t*-test (**c**, **g**), a two-tailed Mann-Whitney test (**a**, **b**, **d**) and two-way ANOVA with Sidak's multiple comparison test (**e**, **f**). Data represent two (**a**–**c**, **e**–**g**) independent experiments or are pooled from three (**d**) independent experiments. Source data for **a**–**g** are provided as a Source Data file.

transcriptomes of naive and activated CD8 T cells (for 12 h in vitro) from WT, let-7 and lin28 transgenic mice (Fig. 4a, b, Supplementary data 2, 3). Based on principle component analysis, the modulation of let-7 expression had a profound impact on the transcriptomes of naive and activated CD8 T cells, potentially influencing the outcome of activation and differentiation. Among the significantly dysregulated DEGs in naive T cells, let-7 suppressed expression of genes involved in cell cycle (*Pcna, Cdc34*), protein synthesis (*Eif4e2, Eif5a, Kars*), and metabolism (*Slc16a10* and *Hk2*), corroborating our previously published results that let-7 is required for the quiescent state of naive CD8 T cells[25]. In antigen-stimulated T cells, in addition to restricting proliferation, let-7 inhibited the expression of genes that control CTL differentiation including transcription factors and their regulators (*Myc, Ezh2, Bhlhe40, Notch2, Rbpj*), pro-survival factor *Bcl2l1*, many genes of cytokine signaling pathways (*Il12rb2, Il4ra, Il2, Jak3, Stat5*) and effector molecules (*Ifng and Gzmc*) (Fig. 4c). These findings were further confirmed by measuring RNA expression of some of the key genes identified (Fig. 4d). Therefore, these results underscored the importance of let-7 expression during early T cell activation, suggesting that let-7 may restrict the strength of TCR-signaling upon antigen recognition. To verify this possibility, we measured Nur77^GFP reporter expression as a proxy of TCR-signaling[81] in naive and activated P14 CD8 T cells isolated from Nur77^GFP, let-7TgNur77^GFP and lin28TgNur77^GFP mice. Although naive T cells did not show any significant difference in GFP expression, activated cells exhibited inverse correlation between GFP intensity and let-7 expression (Supplementary Fig. 8a), confirming the inhibitory role of let-7 in early signaling.

Next, to understand the mechanistic basis of let-7 mediated restriction of TCR-signaling, we analyzed the transcriptomes of naive, activated and differentiated CD8 T cells with different levels of let-7 expression to find dysregulated pathways. Gene set enrichment analysis revealed that the most common pathways inhibited by let-7 were related to mTOR activation, ROS production and metabolic support, including cholesterol homeostasis, glycolysis and oxidative phosphorylation (Fig. 5a, Supplementary Fig. 8b). Indeed, let-7 significantly restricted components of the PI3K/AKT/mTOR pathway including phosphorylation of ERK, AKT and S6 and production of reactive oxygen species (ROS) in in vitro stimulated lymphocytes (Fig. 5b).

Previous reports implicated enhanced activity of the PI3K/AKT/mTOR pathway and overproduction of ROS in promoting terminal differentiation of CD8 T cells and onset of exhaustion, while compromising formation of immunological memory and stemness[61,71,82,83]. Moreover, it has been demonstrated that the key components of the PI3K/AKT/mTOR pathway are direct targets of let-7 miRNAs in different cell types, including T lymphocytes[28,84,85]. Therefore, to assess the contribution of high mTOR activity and ROS production to the terminal effector phenotype of let-7 deficient CTLs, naive lin28Tg CD8 T cells were differentiated into CTLs in the presence of the mTOR

inhibitor rapamycin or the ROS scavenger N-acetyl-L-cysteine (NAC, see Supplementary Fig. 9a for cell viability). Tim3 and CD62L were used as proxy markers for the terminally differentiated and memory phenotypes of CTLs, respectively (Fig. 5c). Both inhibitors lowered the expression of Tim3, PD1, and 2B4; however, the expression of CD62L was rescued only by rapamycin, suggesting that mTOR, and to some extent ROS, are the drivers of the let-7 regulated program in CTLs (Fig. 5c, d, e, Supplementary Fig. 9b). Importantly, mTOR inhibition during the first 48 h of CD8 T cell activation was sufficient to prevent terminal differentiation of lin28Tg CTLs, while rapamycin treatment later during differentiation had almost no impact (Fig. 5f), supporting our previous observation that let-7 expression specifically early during differentiation was most potent in generating CTLs with memory potential. Conversely, NAC had the opposite effect, where ROS inhibition restricted the terminal/effector phenotype of lin28Tg CTLs only at the later time point of differentiation (Fig. 5f). The combination of early rapamycin and late NAC treatments had a mild synergistic effect, suggesting an upstream role for mTOR in controlling ROS generation. Indeed, in vitro activated CD8 T cells in the presence of rapamycin significantly lowered ROS production (Fig. 5g). Next, by crossing P14 lin28Tg with *CD8CreRaptor^fl/fl* or *CD8CreRictor^fl/fl* mice, we determined that specifically TORC2 complex, but not TORC1, is responsible for the terminal/effector phenotype in lin28Tg CTLs (Fig. 5h).

To test whether mTOR/ROS inhibition will rescue the anti-tumor activity of lin28Tg CTLs, we generated P14 lin28Tg CTLs in the presence or absence of rapamycin/NAC treatment and P14 let-7Tg CTLs, and then injected them into B16^gp33 tumor-bearing mice (Fig. 5i). The group of mice that received P14 lin28Tg CTLs with rapamycin/NAC treatment had reduced tumor growth which significantly prolonged mouse survival in comparison to the group with untreated P14 lin28Tg CTLs (Fig. 5j, Supplementary Fig. 9c), suggesting that inhibition of mTOR and ROS partially rescued the cytotoxic function of let-7 deficient CTLs in vivo. The difference in efficiency of let-7Tg and lin28Tg rapamycin/NAC treated CTLs may indicate the contribution of other let-7 controlled mechanisms (Fig. 5a).

To predict the direct let-7 mRNA targets that can be responsible for driving the phenotype of differentiated CTLs, we cross-referenced the list of predicted mouse mRNA targets by TargetScan with DEGs from cluster−1 for naive, activated and differentiated cells in our RNA-seq, where the genes are derepressed in the absence of let-7 and inhibited by the expression of let-7 transgene (Figs. 1e, 4b, Supplementary data 1, 2, 3). Although the let-7 miRNA family has over 1000 predicted mRNA targets in the mouse genome (TargetScan), only a fraction of these genes are expressed in CD8 T cells and were present in cluster-1 (Fig. 5k, Supplementary Data 4). Further investigation will be required to test individual and combinatorial contributions of these genes in the let-7-mediated phenotype of CTLs.

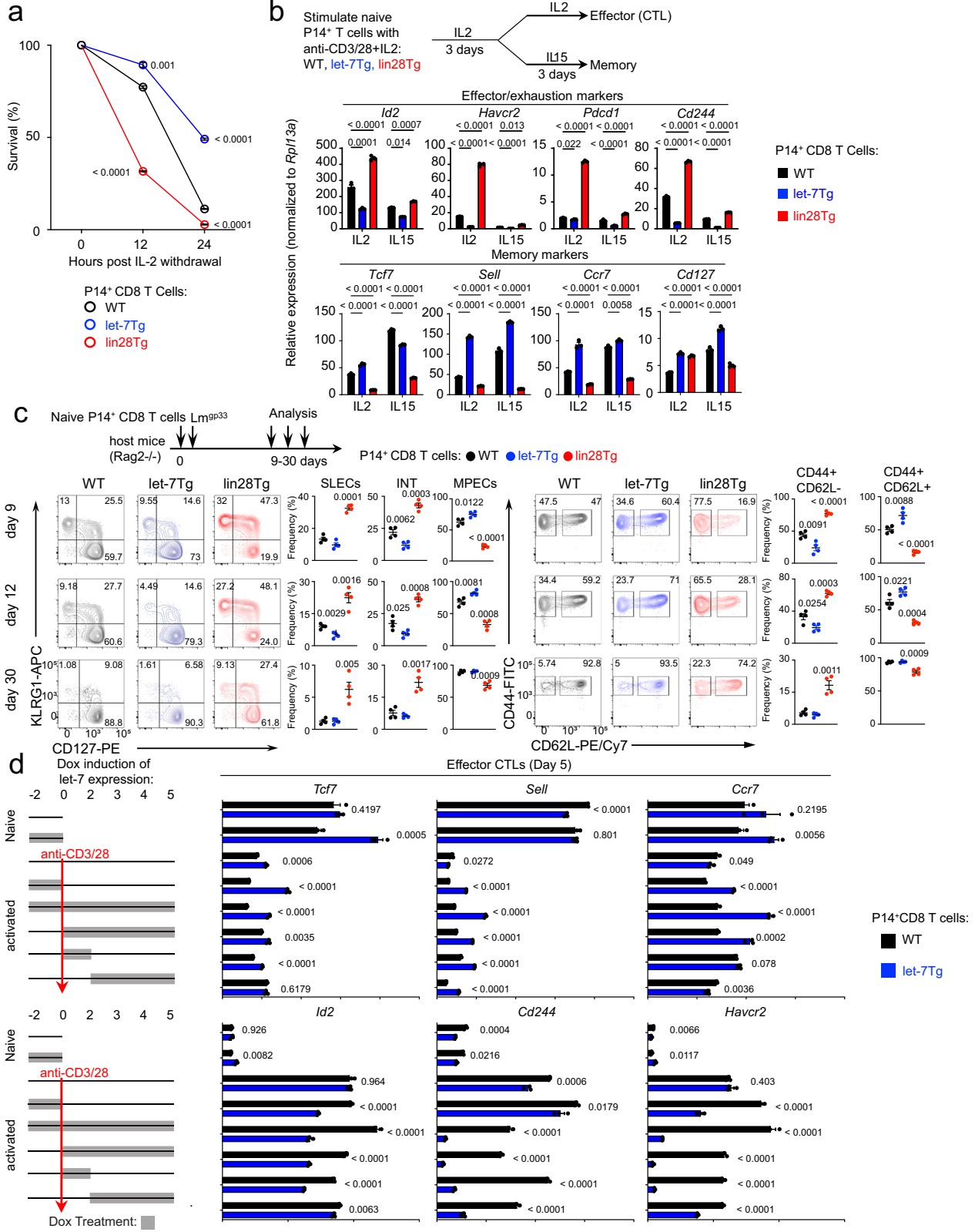

## Discussion

In this study, we investigated the paradox that let-7 deficient CTLs exhibit superior cytotoxic activity in vitro but fail to control tumor growth in vivo, while let-7 transgenic CTLs did the opposite. We demonstrated that retention of let-7 expression during early activation facilitates memory CD8 T cell generation, while downregulation of let-

7 results in the formation of a CTL effector population prone to exhaustion and eventually cell death in the tumor immunosuppressive environment, regardless of stage of activation. Our results, together with previous observations, highlight the importance of early events during T cell activation for the differentiation of memory versus effector CD8 T cells[2,13,76,78,79,86]. Whereas memory potential is developed

**Fig. 3 | Expression of let-7 supports the memory program in differentiating CD8 T cells. a** Cytokine withdrawal assay with P14+ CTLs from either WT, lin28Tg or let-7Tg mice. **b** Experimental design (**b**, top), quantitative RT-PCR analysis of the expression of genes involved in terminal differentiation or memory formation in indicated P14+ CTLs after culture with IL-2 or IL-2 + IL−15. **c** Experimental design (**c**, top), representative FACS of the surface expression of KLRG1 vs CD127 and CD62L vs CD44 with quantification of the frequencies of KLRG1+CD127− (SLECs), KLRG1+CD127+ (INT intermediate), KLRG1−CD127+ (MPECs) populations (left) and CD62L+CD44− and CD62L+CD44+ populations (right) at day 9 and 12 post-*Lm-gp33* infection in the blood and day 30 in the spleen (*n* = 4 per group). **d** Quantitative RT-PCR analysis of the expression of genes involved in terminal differentiation or memory formation in P14+ naive CD8 T cells and P14+ CTLs from WT (black) and let-7Tg (blue) mice with addition of dox at indicated time points (gray bars). Data are the means ± s.e.m. of technical triplicates; *P*-values were determined using a two-tailed unpaired Student's *t*-test. Data represent two (**a**–**d**) independent experiments. Source data for **a**–**d** are provided as a Source Data file.

early and relies on weak TCR-signaling[1,87–90] to maintain high let-7 expression, this trajectory can be diverted at any point by the downregulation of let-7 via strong or persistent TCR-signaling[25], which can occur in tumors or during chronic infection. The significance of the regulation of let-7 expression has also been shown during development of CD8 T cells in early life, where neonatal CD8 T cells with residual expression of fetal lin28b display an antigen-experienced phenotype, and fail to generate responsive memory CD8 T cells[91,92]. Overall, our data offer a strategy in developing next-generation immunotherapies by preserving the multipotency of CTL precursors rather than enhancing the efficacy of differentiation.

Mechanistically, let-7 translates strength and duration of TCR-signaling into metabolic changes mediated by mTOR and consequently production of ROS, both known regulators of CD8 T cell differentiation[61,71,93,94]. Specifically, mTOR inhibition has previously been established to enhance memory CD8 T cell formation. We found that let-7 is able to support memory formation in part by preventing mTOR-mediated hyperactivation. In fact, our data highlight the benefits of mTOR inhibition during early CD8 T cell differentiation by rescuing the terminal differentiation of lin28Tg CTLs in vitro, and enhancing tumor rejection in vivo. Therefore, our results emphasize the importance of the temporal regulation of mTOR activity during CD8 T cell differentiation. This is further supported by the recent observation that the development of the stem-like CD8 T cell pool that feeds the functional CD8 T cell population during PD1 immunotherapy is controlled through a similar mechanism[95]. The roles for mTOR complexes in CD8 T cell differentiation have been previously observed. Consistent with our results, T cell specific mTORC2 deficiency significantly enhanced memory differentiation[96]. However, constitutive mTORC1 activity also led to terminal differentiation at the cost of generating a memory population. Further, mTORC1 deficiency was also insufficient to mount productive recall responses, due to metabolic impairment. These observations highlight the need for a delicate balance of these signals in the context of memory CD8 T cell generation. Although the heightened mTORC2 activity observed in let-7-deficient cells contributes to the diversion of CTLs into an exhaustion-prone terminal effector state, rapamycin, a pan-mTOR inhibitor, was more potent for redirecting the differentiation of lin28Tg CTLs into a memory prone state than individual knockouts of TORC1 or TORC2, suggesting a compensatory mechanism of mTOR complexes. Accordingly, fine tuning of the activity of both mTOR complexes, for which miRNAs such as let-7 are exquisitely designed, is imperative for functional memory generation. This has important implications for the use of non-specific and constitutive mTOR inhibitors, such as rapamycin, in the clinic.

Despite the profound impact of let-7 miRNA modulation on the function and identity of CD8 T cells, the use of let-7 as an immunotherapeutic tool has only been explored from the perspective of the tumor cell. Let-7 expression is often downregulated in tumors, due to aberrant expression of Lin28, and leading to derepression of PDL1[97,98]. However, the majority of therapies attempting to rescue let-7 expression in tumor cells are delivered systemically[99]. Moreover, elevated expression of let-7e in myeloid cells is correlated with poor immunotherapy responses, demonstrating that high let-7 expression among immune cell subtypes does not provide the same advantages[100]. Accordingly, targeted delivery of let-7 into CD8 T cells will be important for the efficacy of this proposed therapeutic strategy. Specifically, the upregulation of let-7 expression during CAR-T cell generation would provide a unique opportunity to introduce let-7 in a CD8 T cell-intrinsic manner. Such an approach could also help to resolve onset of exhaustion, and the lack of long-lasting memory CAR-T cells in this treatment regimen[101–103].

Taken in the context of these previous findings, our work contributes to the identification of a universal program controlled by let-7, where let-7 must be appropriately regulated to mount effective CD8 T cell responses. Ultimately, this study identified let-7 miRNAs as a therapeutic tool of great potential interest, where both its overexpression and depletion, impact distinct components of CD8 T cell immunity.

## Methods

### Ethics statement

This study was performed in accordance with the recommendations in the Guide for the Care and Use of Laboratory Animals of the National Institutes of Health. All animals were handled according to approved institutional animal care and use committee (IACUC) protocols (#2186, 2955) of the University of Massachusetts.

### Animals

C57BL/6 J (CD45.2+ wild type, stock no. 000664), B6.SJL- *Ptprc^aPepc^b*/BoyJ (CD45.1+ wild type, stock no. 002014), B6.Cg-*Rag2^tm1.1Cgn*/J (*Rag2^−/−*, stock no. 008449), Tg(tetO-cre)1Jaw/J (iCre, stock no. 006224) and C57BL/6-Tg(Nr4a1-EGFP/cre)820Khog/J (Nur77^GFP^, stock no. 016617) mice were obtained from the Jackson Laboratory. Gzmb^Cre+^ (B6;FVB-Tg(GZMB-cre)1Jcb/J) mice were a generous gift from Dr. Rodriguez (Moffit Cancer Center). R26^STOP-Lin28-GFP^ mice were generated in Dr. Singer's lab (NCI/NIH) in collaboration with Dr. Hollander (University of Oxford). P14+lin28Tg and P14+let-7Tg mice on a *Rag2^−/−*background were described previously (eLife). Gzmb^Cre+^R26^STOP-Lin28-GFP^ mice were generated by crossing R26^STOP-Lin28-GFP^ mice with Gzmb^Cre+^ mice. iCre R26^STOP-Lin28-GFP^ mice were generated by crossing R26^STOP-Lin28-GFP^ mice with iCre. WTNur77^GFP^, let-7TgNur77^GFP^ and lin28TgNur77^GFP^ mice were generated by crossing P14 + WT, P14+lin28Tg and P14+let-7Tg mice on a *Rag2^−/−*background to Nur77^GFP^ mice. Both female and male 6−10-week-old mice were used for all experiments. Animals were maintained in ventilated cages under specific pathogen-free conditions and a 12 h dark/12 h light cycle at 20−22 °C and a humidity range of 30−70%. Animals were fed with irradiated chow from LabDiet (standard diet cat# 5P76, breeders deit Cat# 5058). All breedings were maintained at the University of Massachusetts, Amherst.

### Doxycycline-mediated induction of let-7 transgene expression

Experimental mice including control animals (unless specifically stated otherwise) were fed with 2 mg/mL doxycycline (Sigma, Cat# D9891) in drinking water supplemented with 10 mg/mL sucrose (Sigma, Cat# S9378) for four days prior to the initiation of experimental procedures to ensure maximal induction of let-7g expression. Doxycycline drinking water was replaced every other day. In vitro, lymphocytes were cultured with 2 µg/mL doxycycline in culture media (see in vitro culture below).

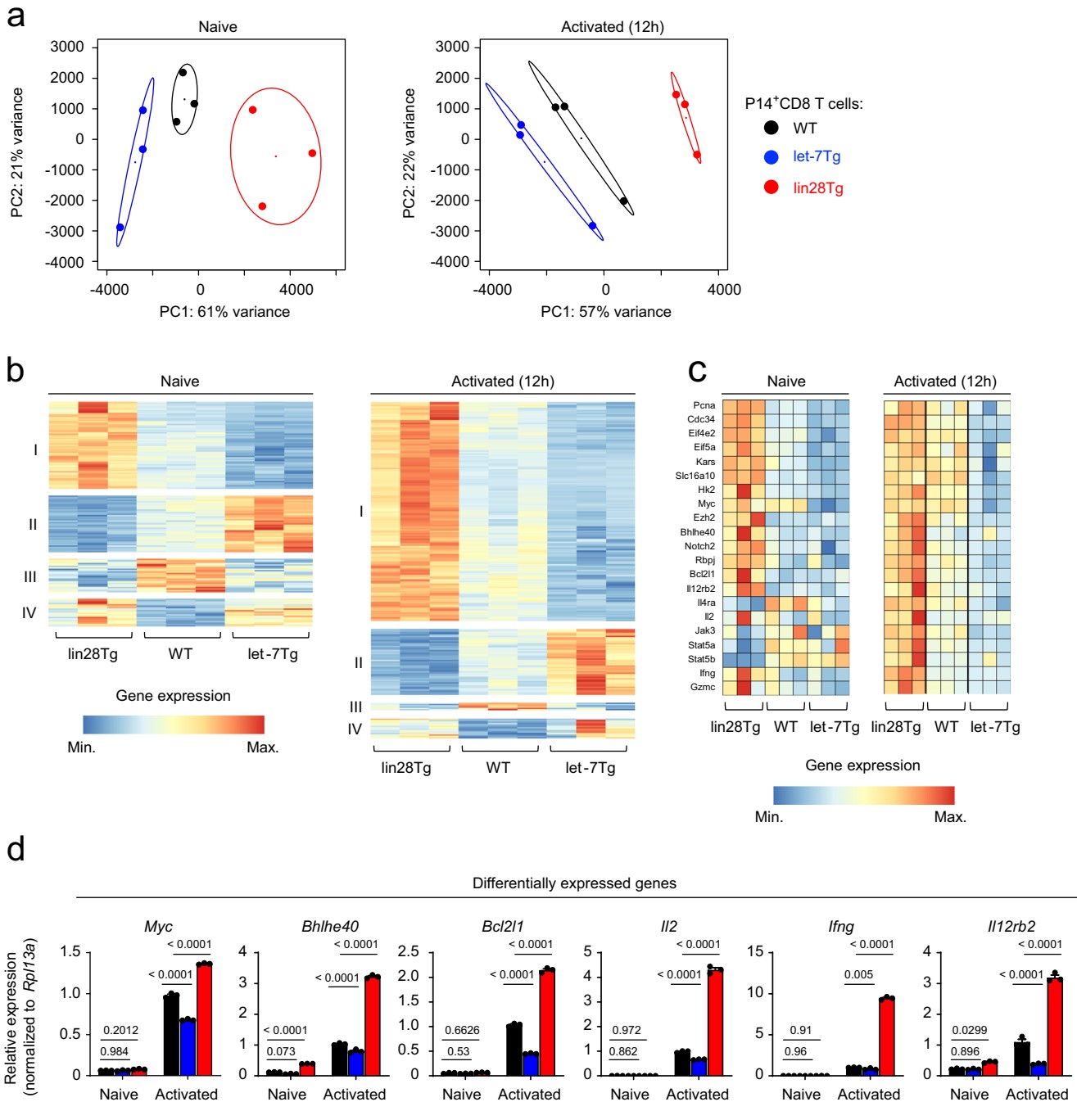

Fig. 4 | The impact of let-7 expression on the transcriptomics of naive and activated CD8 T cells. a–c, RNA-seq of naive and activated (12 h with anti-CD3/anti-CD28) P14+ CD8 T cells from either WT, let-7Tg or lin28Tg mice. Data are from one RNA-seq analysis comprised of three biological replicates per group. a Principal component analysis of all DEGs. b Cluster analysis of upregulated and down-regulated DEGs. c Heatmaps of selected DEGs in naive and 12h-activated WT, let-7Tg or lin28Tg P14+ CD8 T cells. d Quantitative RT-PCR analysis of the expression of indicated genes from (c). Data are the means ± s.e.m. of technical triplicates (d); P-values were determined using a two-tailed unpaired Student's t-test. Data represent two (d) independent experiments. Source data for d are provided as a Source Data file.

## Flow cytometry (FACS) analysis

Flow cytometry data were acquired on a LSRFortessa and analyzed with FlowJo software. For gating strategy see Supplementary Fig. 10. Dead cells were excluded by staining with 4′,6-diamidino-2-pheny-lindole, dilactate (DAPI, Biolegend, Cat# 422801). The following monoclonal antibodies from BioLegend were used: CD8α-Pacific Blue (1:20 dilution, clone 53-6.7, Cat#100725), CD8α-APC/Cy7 (1:20 dilution, clone 53-6.7, Cat# 100714), CD8α-APC (1:200 dilution, clone 53-

6.7, Cat# 100712), CD4-Pacific Blue (1:20 dilution, clone RM4-5, Cat# 100531), CD44-FITC (1:40 dilution, clone IM7, Cat#103021), CD62L-PE/Cy7 (1:50 dilution, clone MEL-14, Cat#104418), KLRG1-APC (1:10 dilution, clone 2F1/KLRG1, Cat#138412), CD127-bio (1:10 dilution, clone A7R34, Cat#135006), PD1-bio (1:10 dilution, clone 29 F.1A12, Cat# 135212), Tim3 (1:10 dilution, clone RMT3-23, Cat# 119704), 2B4-bio (1:10 dilution, clone m2B4(B6)458.1, Cat# 133506), CD38-PE/Cy7 (1:10 dilution, clone 90, Cat# 102718), CD39-APC (1:40 dilution, clone

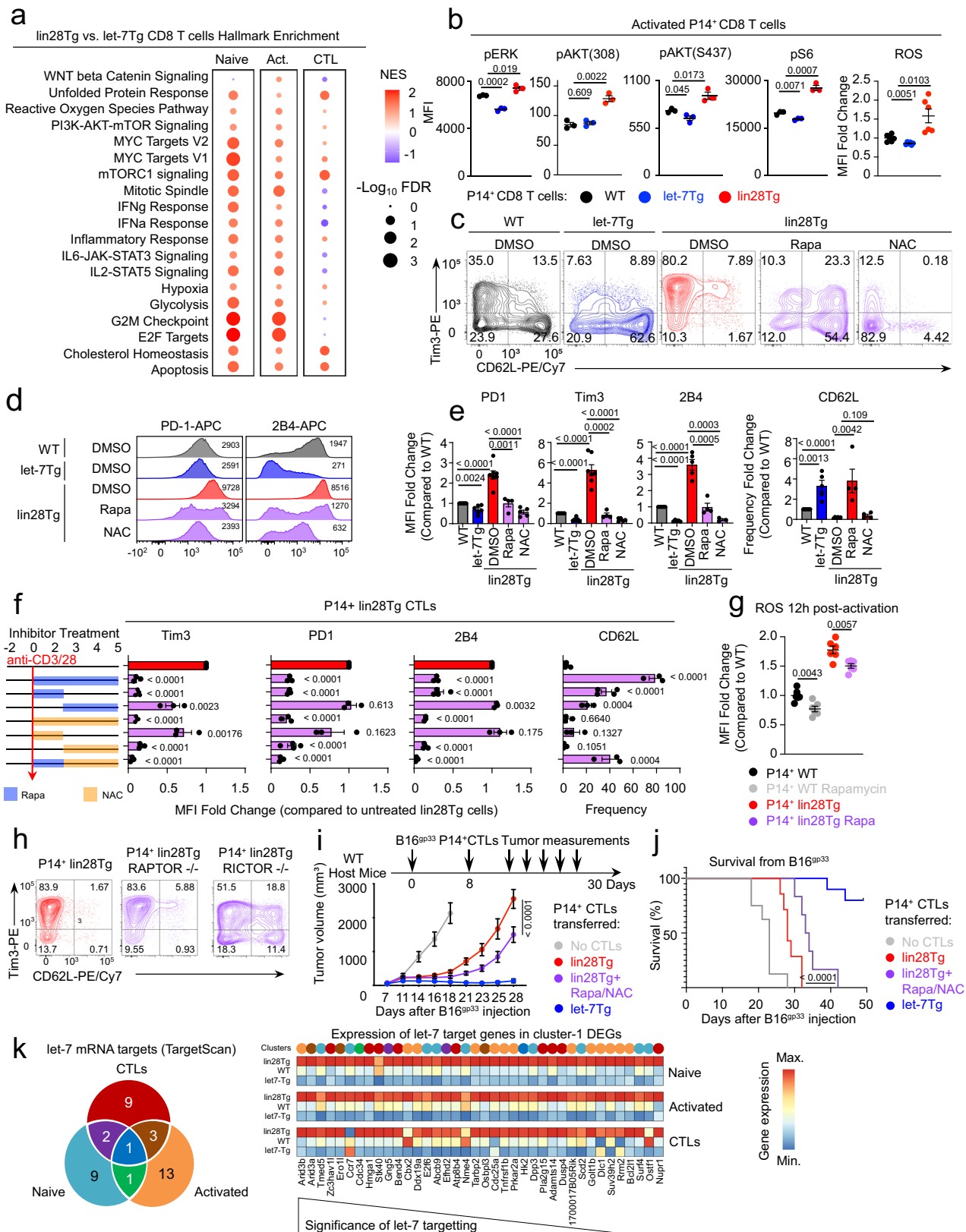

Duha59, Cat# 143810), CD45.2-FITC (1:20 dilution, clone 104, Cat#109806), CD45.2-PE (1:20 dilution, clone 104, Cat#109808), CD45-PE/Cy7 (1:200 dilution, clone 30-F11, Cat#103114), phospho-Erk-PE (1:10 dilution, clone 6B8B69, Cat# 369505), IL-10-PE (1:10 dilution, clone JES5-16E3, Cat# 505008), CD27-APC (1:10 dilution, clone LG.3A10, Cat# 505008), CXCR3-BV650 (1:10 dilution, clone CXCR3-

173, Cat#126531), TNFα-PE/Cy7 (1:20 dilution, clone MP6-XT22, Cat# 506324). CD160-PE (1:10 dilution, clone CNX46-3, Cat# 12-1601-81) and IFNγ-APC (1:100 dilution, clone XMG 1.2, Cat# 17-7311-81) were from eBioscience. CD28-PE (1:10 dilution, clone 37.51, Cat# 553297) and TCF-1-AF647 (1:10 dilution, clone S33-966, Cat# 566693) were from BD Biosciences. Phospho-S6-PE (1:10 dilution, clone D57.2.2E, Cat# 5316),

**Fig. 5 | Let-7 controls CTL differentiation through inhibition of mTOR/ROS pathways. a** Gene set enrichment analysis of top hallmark pathways upregulated and downregulated in P14[+] naive, 12h-activated and 5-day differentiated CD8 T cells from lin28Tg mice in comparison to those from let-7Tg mice. RNA-seq data are from Figs. 1 and 4. **b** MFIs of phospho-Erk, phospho-Akt (Thr308), phospho-Akt (S473), phospho-S6 and ROS in P14[+] CD8 T cells from WT, lin28Tg or let-7Tg mice activated in vitro ($n = 3$ per group). **c, d** Representative FACS analysis of expression of indicated markers in P14[+] CTLs from WT, let-7Tg or lin28Tg mice differentiated with rapamycin (rapa) or NAC. **e** MFIs of indicated proteins and frequencies of CD62L-positive cells presented as a fold change relative to WT. **f** MFIs of indicated proteins and frequency of CD62L-positive cells in P14[+] CTLs from lin28Tg mice differentiated with rapamycin or NAC added at indicated time points. Data are presented as a fold change relative to untreated control. **g** MFIs of ROS in P14[+] CD8 T cells from WT or lin28Tg mice activated with or without rapamycin.

**h** Representative FACS analysis of Tim3 and CD62L expression in P14[+] CTLs from lin28Tg mice on Raptor-/- or Rictor-/- background. **i, j** Experimental design, tumor growth curves (**i**) and Kaplan–Meier survival curves (**j**) of WT mice s.c. injected with B16[gp33] tumor cells and adoptively transferred with P14[+] CTLs from lin28Tg mice differentiated with ($n = 8$ mice) or without ($n = 7$ mice) rapamycin+NAC treatment or let-7Tg mice ($n = 10$ mice). For control group that received no CTLs, $n = 8$ mice. **k** Venn diagram showing the number of let-7 target genes in P14[+] naive, 12h-activated and 5-day differentiated CD8 T cells from cluster−1 of Fig. 1e and heatmap displaying the expression of these genes. The numbers in **c, h** indicate the frequency of populations in each quadrant. The numbers in **d** indicate MFIs. *P*-values were determined using a two-tailed unpaired Student's *t*-test (**b, e–g**), c (**i**) and log-rank Mantel-Cox test (**j**). Data represent three (**b–d, h**) and two (**i, j**) independent experiments, or are pooled from three (**e, f**) and two (**g**) independent experiments. Source data for **b, e–g, i, j** are provided as a Source Data file.

phospho-Akt-PE (1:10 dilution, clone D25E6, Cat# 13842), phospho-Akt-APC (1:10 dilution, clone D9E, Cat# 11962) and Foxo1-PE (1:10 dilution, clone C29H4, Cat# 14262) were from Cell Signaling Technology. Streptavidin-AF647 (1:40 dilution, Cat# 405237) and Streptavidin-PE (1:40 dilution, Cat# 405204) were from Biolegend.

Live cells were treated with α-CD16/32 Fc block (1:40 dilution, clone 4G2, Cat# 553142, BD Biosciences) prior to staining with antibodies against surface markers. Staining for surface proteins was performed at 4 °C for 40 min and FACS buffer (PBS + 0.5% BSA + 0.01% sodium azide) was used for washes.

For intracellular cytokine staining, cell suspensions were restimulated in vitro for a total of 4 h with 50 ng/mL phorbol 12-myristate 13-acetate (PMA, Sigma, Cat# 524400) and 1 μM Ionomycin (Sigma, Cat# 407952) with addition of 2 μM monensin (eBioscience, Cat# 00-4505-51) to inhibit secretion. After surface antibody staining, cells were stained with the Live/Dead fixable Aqua Dead Cell Stain Kit (Thermo-Fisher Scientific, Cat# L34957) according to the manufacturer's instructions. Then cells were fixed and permeabilized using the Cyto-fix/Cytoperm solution kit (BD Biosciences, Cat# 554714) according to the manufacturer's instructions, followed by staining with antibodies against intracellular molecules. For transcription factor staining, Foxp3/ Transcription factor staining buffer set (eBioscience, Cat# 00-5523-00) was used and staining was performed according to the manufacturer's instructions.

For phospho-S6 staining, cells were fixed in 2% paraformaldehyde for 20 min at 37 °C, washed twice with PBS/10%FBS, fixed with ice-cold methanol/acetone (1:1) for 20 min at −20 °C, washed twice with PBS/10%FBS and stained. For phospho-Erk and phospho-Akt staining, cells were fixed in 2% paraformaldehyde for 20 min at 37 °C, washed twice with PBS/10%FBS, fixed with ice-cold 90% methanol for 20 min at −20 °C, washed twice with PBS/10%FBS and stained. Staining for ROS was performed using CM-H2DCFDA (ThermoFisher Scientific, Cat# C6827) at the final concentration of 1uM according to manufacturer's instructions. Staining for mitochondrial potential was performed with JC-1 (Sigma, Cat# T3168) at the final concentration of 2uM and TMRE (Sigma, Cat# 87917) at the final concentration of 20 nM in culture media at 37 °C, 5% $CO_2$ for 30 min.

## T cell culture

Lymph nodes and spleens were harvested and gently tweezed to remove lymphocytes. Spleen cells were then lysed using ACK lysing buffer (KD Medical, Cat# RGC-3015) to remove erythrocytes. For CTL generation, cells were stimulated with irradiated splenocytes and soluble anti-CD3 mAbs (2 μg/mL). Cells were cultured in RPMI supplemented with 10% fetal bovine serum, 1% HEPES, 1% sodium pyruvate, 1% penicillin/ streptomycin, 1% L-glutamine, 1% non-essential amino acids and 0.3% β-mercaptoethanol. 48 h after activation IL-2 was added to culture media (100 U/mL) and cells were differentiated for additional 3 days; 2 μg/mL doxycycline was added when necessary. For cytokine withdrawal assay, differentiated CTLs were washed twice,

counted, and $5 \times 10^4$ CTLs were plated in media without IL-2. Cell viability was assessed at 12 h and 24 h by flow cytometry using propidium iodide. For memory formation, differentiated CTLs were washed, plated with media containing IL-15 (50 U/ml) or IL-7 (1 ng/ml) and cultured for additional 3 days. For pERK, pAKT, pS6 and ROS staining, cells were activated with plate-bound anti-CD3 mAbs (1 μg/mL) and anti-CD28 mAbs (5 μg/mL) for 30 min (pErk, pAkt), 1 h (pS6) or 20 h (ROS). For tumor cytotoxicity assay titrated amounts of CTLs were added to $1 \times 10^6$ B16[gp33] adherent tumor cells plated one day prior. Cells were co-cultured for 12–16 h. Cytotoxicity was assessed by propidium iodide in GFP+ tumor cells by flow cytometry. For CTL differentiation with inhibitors, rapamycin (Sigma, Cat# 553211) was used at the final concentration of 0.5uM, NAC (Sigma, Cat# A1965) was used at the final concentration of 10 mM.

## Experiments with Listeria monocytogenes (Lm-gp33)

Recombinant *Lm*-gp33 strain expressing GP(33-41) epitope was a gift from Dr. Harty (University of Iowa). Experimental stocks were stored at −80 °C. Frozen aliquots of *Lm*-gp33 were thawed, cultured in Tryptic Soy Broth media (KD Medical, Cat# CUS-0279) containing 50 ug/ml streptomycin and the concentration was measured by optical density at 600 nm wavelength (OD600) every hour. Once OD600 was 0.08-0.09, cells were spun down for 15 min at 6000 RPM, 4 °C, resuspended in 0.9% NaCl at the concentration of $6 \times 10^7$ CFU/ml (OD600 of 1 refers to $1 \times 10^9$ CFU of *Lm*-gp33). Each recipient mouse was infected intravenously (i.v.) with $6 \times 10^6$ CFU of *Lm*-gp33. The same day prior to infection mice were injected i.v. with $2 \times 10^4$ naive P14[+] CD8 T cells.

## Retroviral vectors and transduction

Retroviruses were produced from the transfection of Platinum-E cells with empty and *Bcl2l1*-expressing pMRX-IRES-GFP vectors using Lipofectamine 2000 (Life Technologies, Cat# 11668019) according to manufacturer's instructions. Viral supernatants were collected 24 h after transfection, concentrated with PEG-it (System Biosciences, Cat# LV825A-1) and frozen in aliquots. For retroviral transduction, naive lymphocytes were stimulated with irradiated splenocytes in the presence of anti-CD3 mAbs (2 μg/mL) for 14 h and then spinfected (2000 RPM, 90 min, 37 °C) with virus and polybrene (4 μg/mL, Sigma, Cat# TR-1003-G). Media was changed 4 h after transduction, at this point IL-2 was added at 100 U/mL and CTLs were generated.

## Assessment of metabolism

Extracellular acidification rate (ECAR) and oxygen consumption rate (OCR) were measured using the Glycolytic Rate and Cell Mito Stress Assay Kits (Agilent, Cat# 103344-100 and 103015-100, respectively) according to manufacturer's instructions. Briefly, CTLs were resuspended in RPMI containing 1 mM HEPES, 1 mM pyruvate, 2 mM glutamine and 10 mM glucose and plated onto Seahorse cell plates ($1 \times 10^5$ cells per well) coated with Cell-Tak (Corning, Cat# 354240). ECAR and OCR were then measured using a Seahorse XFe96 Extracellular Flux

Analyzer (Agilent) under basal conditions and in response to rotenone/antimycin (0.5 µM) and 2-deoxy-D-glucose (50 mM) for ECAR or in response to oligomycin (50 µM), FCCP (50 µM) and rotenone/antimycin (25 µM) for OCR.

## Tumor cell lines

B16-F10 melanoma cells (ATCC, CRL-6475), MC57 fibrosarcoma cells (ATCC, CRL-2295) and EL4 thymoma cells (ATCC, TIB-39) were obtained from ATCC. B16-F10 and MC57 were cultured in DMEM supplemented with 10% FBS, EL4 cells were cultured in RPMI supplemented with 10% FBS. To generate B16$^{gp33}$ and EL4$^{gp33}$, B16-F10 and EL4 were transduced with pHRST-IRES-eGFP lentiviral vector modified to express gp33 peptide, and FACS-sorted for GFP-positive population.

## Tumor experiments

Mice were given a sublethal dose of irradiation (500 Rad) and then injected subcutaneously in the right flank with $2.5 \times 10^5$ B16 or $2.5 \times 10^5$ EL4 or $2 \times 10^6$ MC57 cells. For studies involving adoptive T cell transfer, $2 \times 10^6$ P14+ CTLs were injected i.v. into mice once tumors were established (at day 8 after tumor inoculation). For studies involving checkpoint blockade, mice were given 200 ug of anti-PDL1 or isotype control (BioXCell, Cat# BE0101and BE0090, respectively) antibodies intraperitoneally at days 8, 11, 14 and 17. For studies involving let-7Tg mice, all mice were fed with doxycycline for the duration of the study. Tumors were measured every 2–3 days with a caliper and tumor volume was determined using the following formula: $\frac{1}{2} \times D \times d^2$ where D is the large diameter and d is the small diameter. Mice were euthanized when tumors reached 2 cm³ or when tumors became ulcerated, determined by daily inspection and approved by UMass IACUC protocol. In instances when this limit was exceeded, veterinarians were consulted, and mice were only kept until the end of the experiment if they did not show signs of distress and tumors did not interfere with normal behavior.

For TIL isolation, tumors were minced and digested with 1 mg/mL collagenase D and 200 µg/mL DNase mix (Sigma, Cat# 11088882001 and 10104159001, respectively) by incubating at 37 °C for 30 min with mixing every 5 min. The digestants were then passed through 40-micron filter, spun at 1250 rpm for 5 min and resuspended in FACS buffer.

## Isolation of RNA and quantitative PCR

RNA was isolated using the Total RNA Purification Kit and genomic DNA removed using the RNase-Free DNase I Kit (Norgen Biotek, Cat# 37500 and 25710, respectively). cDNA was synthesized using the SensiFast cDNA Synthesis Kit (Thomas Scientific, Cat# C755H66). SYBR Green quantitative PCR was performed using the SensiFAST SYBR Lo-Rox kit (Thomas Scientific, Cat# C755H95) and Taqman quantitative PCR was performed using the SensiFAST Probe Lo-Rox kit (Thomas Scientific, Cat# C755H88). Both SYBR Green and Taqman amplification primers (Integrated DNA Technologies, or Applied Biosystems) are listed in Supplementary data 5.

## RNA-seq

$20 \times 10^6$ in vitro-generated CTLs and at least $4 \times 10^6$ naive and 12 h-activated CD8 + T cells were lysed in TRIzol (ThermoFisher Scientific, Cat# 15596026) and sent to Novogene for RNA extraction, library preparation, sequencing and initial data processing. Reference genome and genome annotation files were downloaded from genome website browser (NCBI/UCSC/Ensembl) and indexed using Bowtie v2.0.6. Clean, paired-end reads were aligned to the reference genome with TopHat v2.0.9. Read counts per gene and FPKM were determined using HTSeq v0.6.1. Differential gene expression analysis between the WT, Let-7Tg, Lin28Tg sample groups ($n = 3$ in each group) was performed using the DESeq2 R package (v2_1.6.3). Adjusted p-values were calculated with the Benjamini-Hochberg multiple test correction, and

genes with an adjusted $p$-value < 0.05 were considered significantly differentially expressed between two groups.

## Analysis of RNA-seq data

Principal component analysis (PCA) was performed with normalized read counts using the PCAtools (v2.8.0), car (v3.1-2), RColorBrewer (v1.1-3), and ggplot2 (v3.4.2) R packages. Volcano plots were generated with differential gene expression data using the EnhancedVolcano (v1.14.0) R package. Heatmaps were generated with normalized read counts using the pheatmap (v1.0.12), dendextend (v1.17.1), and RColorBrewer (v1.1-3) R packages. Differential expression of each gene was ranked according to the adjusted p-value and log2 fold change. Gene sets for memory and terminal effector markers were obtained from MSigDB[74,104]. Hallmark gene sets obtained from MSigDB were converted to mouse ortholog genes[105]. The GSEA software[106,107] version 4.2.3 (Broad Institute) was used to analyze enrichment of Hallmark features using the GSEAPreranked method.

## Prediction of let-7 miRNA targeted genes

Predicted target genes of let-7 miRNA were compiled from the TargetScan database[108] version 8.0. Differentially expressed genes in the Cluster-1 category of heatmaps (genes significantly upregulated in lin28Tg but significantly downregulated in let-7Tg mice) for each time point were searched for the presence of let-7 miRNA targets. The number of unique or shared target genes in each time point were represented with a Venn diagram.

## Statistical analysis

Data statistical analysis was performed with Prism 9 (GraphPad software). *P*-values were determined using a two-tailed unpaired Student's t-test, a two-tailed Mann-Whitney test, a two-way ANOVA or log-rank Mantel-Cox test as indicated in the figure legends. Sample size was not predetermined by statistical methods, and no data were excluded from the analyses.

## Reporting summary

Further information on research design is available in the Nature Portfolio Reporting Summary linked to this article.

# Data availability

The sequencing data that support the findings of this study have been deposited in the National Center of Biotechnology Information Gene Expression Omnibus (GEO) and are accessible through the accession number GSE232541. The gene sets used for GSEA in this study are accessible from MSigDB under GEO accession codes GSE8678 (GSE8678_IL7R_LOW_VS_HIGH_EFF_CD8_TCELL_UP (gsea-msigdb.org)) and GSE10239 (GSE10239_MEMORY_VS_DAY4.5_EFF_CD8_TCELL_UP (gsea-msigdb.org)). The list of target genes by miRNA were obtained from the TargetScan database (TargetScanMouse 8.0). Source data are provided as a Source Data file. Source data are provided with this paper.

# Code availability

No custom code was used in the study.

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

## Acknowledgements
This work was supported by NIH grants AI146188 (L.A.P.) AI133041 (L.A.P.) and Biotechnology Training Program (BTP) of National Research Service Award T32 GM13096 (K.A.H.). We thank Dr. Barbara A. Osborne and Dr. Richard A. Goldsby for critical reading of the manuscript; Dr. Alfred Singer for providing reagents, and Lin28Tg and P14 mice; the Tetramer Core Facility of the US National Institutes of Health for tetramer reagents.

## Author contributions
A.C.W., E.L.P. and L.A.P. designed the study. A.C.W., K.A.H., C.C.A., J.U., A.C.L., D.J.R., X.L., I.T., A.C., J.M. and E.L.P. performed experiments. S.Zh., S.Z. and G.A.H. designed and generated the mouse model for this study. K.A.H. and C.C.A. provided expertize on data analysis. E.L.P. and L.A.P. supervised the study. A.C.W., E.L.P. and L.A.P. wrote the manuscript.

## Competing interests
The authors declare no competing interests.
