## [Peer Review File · Nature Communications]

Let-7 enhances murine anti-tumor CD8 T cell responses by promoting memory and antagonizing terminal differentiationREVIEWER COMMENTS

Reviewer #1 (Remarks to the Author):

The study by Wells et al describes an in-depth analysis of the role of let-7 miRNAs in mouse in vivo models. The authors performed a series of in-depth analysis of let-7 overexpressing and deficient CD8 T cells with cutting-edge technologies and mouse models. The authors discovered that let-7 deficiency, despite leading to a higher cytotoxic profile in vitro, results into subordinate tumor immunity while let-7 overexpression is superior. The in-depth studies showed that this is related to the induction of an exhaustion phenotype in case of let-7 overexpressing and a memory phenotype in case of let-7 overexpression, which relates the regulation of the mTOR pathway by let-7. Some comments/suggestions remain.

1. The experiment with anti-PDL1 (Fig 2D) is interesting but the tumor model shown in Figure 2d seems unresponsive to PD-1/PD-L1 blockade. Therefore it would be informative to know whether let-7 deficient CD8 T (lin28Tg) can also become less exhausted in a model that is sensitive to blockade of the PD-1/PD-L1 pathway (e.g. MC38), reflecting a clinical situation in which patients respond to PD-1/PD-L1 blockade. In addition, it would be of interest to know whether let-7 overexpressing CD8 T cells (let-7Tg) can also benefit from blocking the PD-1/PD-L1 pathway, in such models. These experiments are of interest for the potential of targeting let-7 miRNAs as a therapeutic tool (see also point 4).

2. The functionality of the Let-7 proficient and deficient CD8 T cells has not been addressed with respect to their polyfunctional cytokine profile (e.g. IFN-g, TNF, IL-2) at the protein level, which is considered as a hallmark of T cell (dys)functionality, especially in the context of exhaustion. The higher gene expression of IL-2 in lin28Tg (Fig 4d) seems unexpected in this respect.

3. The discussion is quite short and it would be of interest if the authors would add discussion on let-7 miRNAs as a therapeutic tool as suggested by the authors (lines 339,340).

Other comments

4. It would be informative to indicate that the authors use mouse in vivo models, and accordingly this should be indicated in abstract and title.

5. Typo: ACKNOWLEDGMENTS > Acknowledgements

Reviewer #2 (Remarks to the Author):

In this study, Wells et al. follow up on prior observations of Let-7 regulating CD8+ T cell differentiation and function in vitro now in vivo. They analyze the contribution of let-7 miRNAs by examining let-7 deficient (lin28Tg) or overexpressing (let-7Tg) CTLs. Let-7 is known to be highly expressed by naïve CD8+ T cells to ensure quiescence and survival and is downregulated during transition into functional effector CD8+ T cells. The authors tested the anti-tumoral capacities of CTLs with different levels of let-7 expression injected into tumor-bearing mice and observed superior tumor control and survival of mice receiving let-7Tg CTLs. RNA-seq analysis of lin28Tg- and let-7Tg CTLs revealed divergent transcriptome profiles, with lin28Tg CTLs showing enhanced expression of effector/exhaustion-associated genes, while let-7Tg CTLs exhibited a memory-like program. In addition, lin28Tg CD8+ T cells exhibited increased mTOR signaling and ROS, which the authors followed up by

performing FACS analysis of phosphorylated ERK, AKT, S6 proteins and ROS production. FACS analyses confirmed these findings of increased pERK, pAKT, pS6 and ROS content in lin28Tg CD8+ T cells compared to let-7Tg CD8+ T cells.

This study highlights the link between let-7 and the regulation of the quiescent naïve/memory program of CD8+ T cells and adds an investigation of a potential mechanism via suppression of the mTOR pathway. However, the authors do not show the molecular mechanism of how let-7 directly promotes the memory program and inhibits the effector program and how it regulates individual components of the mTOR signaling pathway.

The main criticism to the work is therefore that a lot of the effects of let-7 connected to the naïve/quiescent/memory T cell program are known. The conceptual advance here would be the mechanistic investigation regarding the metabolic signaling. However it is well known that naïve, memory and effector and exhausted T cell have different metabolic activity. The issue is that while let-7 can have multiple regulated targets in the metabolic signaling pathways, and this could translate into lower protein expression (not shown for most markers) if this would directly result in lower pathway activity is unclear. It is well possible from the data presented that let-7 could primarily act on regulating major differentiation cues (ie by regulating transcription factors) instead of directly modulating metabolic pathways. This question of direct let-7 involvement would be better addressed in an inducible let-7 system on different differentiation subsets.

Additional Points:

- The authors performed FACS staining and qPCR to validate the different programs of lin28Tg- and let-7Tg CTLs but provide only exemplary histograms for Tim-3, PD-1, CD39, CD160, 2B4, CD38, CD62L and CD127. In this regard, they should expand the data set of this part to draw conclusions based on cellular protein expression. In addition, other markers such as CD57, T-bet, Eomes, KLRG1, Blimp1, CD28, CD27, Foxo1, Bcl-6, CXCR3, STAT3 and STAT4 would also be informative for distinguishing between terminal differentiation vs. memory phenotype.
- The authors note that lin28Tg CTLs exhibit a “terminally differentiated” phenotype. However, they respond to PD-1 checkpoint blockade (see Figure 2d), which seems contradictory since terminally differentiated CD8+ T cells are described as refractory to PD-1 treatment. Is the difference (lin28Tg vs. lin28Tg + aPDL1) in tumor volume statistically significant? Could the authors also provide the data for mice with EL-4gp33 tumors that received WT- or let-7Tg CTLs?
- Can the authors please explain why they observe differences in Id2, Cd244, Tcf7, Sell and Ccr7 expression in naïve CD8+ T cells and differences in Havcr2, Pdcd1, Tcf7, Sell and Ccr7 expression in untreated P14+ CTLs isolated from WT vs. iCre+R26STOP-lin28-GFP mice (Figure 2f)? Because these mice already exhibit basal differential gene expression, they may not be a suitable model to study the time kinetics of lin28 induction.
- The authors demonstrate in Figure 3b that lin28Tg CD8+ T cells are more susceptible to IL-2 treatment, which triggers subsequent upregulation of exhaustion markers, whereas let-7Tg CD8+ T cells are more responsive to IL-15 intervention, which in turn leads to increased expression of memory markers. Did the authors also have the opportunity to examine the effect of IL-7 supplementation on let-7Tg- and lin28Tg CD8+ T cells, given that IL-7, along with IL-15, represents a hallmark cytokine in the formation and maintenance of memory CD8+ T cells?
- It is not intuitive to the reader of the manuscript why the authors frequently change markers indicating terminally differentiated vs. memory CD8+ T cells (e.g. Tim-3/PD-1,

KLRG1/CD127, Tim-3/CD62L axes).

- The authors often make statements about T-cell function being affected by knockdown or overexpression of let-7, but only provide the gene expression of effector genes. It would be more convincing if the authors could measure cytokine production (e.g. IFN, TNF, IL-2) or secretion of granzymes and perforins of stimulated let-7Tg- and lin28Tg CD8+ T cells by FACS analysis.
- By RNA-seq analyses of naïve, 12h-activated, and 5-day-differentiated CD8+ T cells from lin28 or let-7Tg mice, the authors demonstrated a lin28Tg-driven induction of mTOR signaling and the ROS pathway (Figure 5a). To confirm these results, they performed FACS analyses and found increased levels of pERK, pAKT, pS6 and ROS in lin28Tg CD8+ T cells. To underline these interesting findings, the authors should measure MEK and PI3K with respect to mTOR signaling. Because lin28Tg CTLs (vs. let-7Tg CTLs) also showed upregulation of oxidative phosphorylation (Supplementary Figure 8a), the authors should additionally analyze mitochondrial bioenergetics, e.g. by staining MTG, MTD, GSH, JC-1 or TMRE.
- Subsequently, the authors show that mTOR inhibition by rapamycin reverses the exhausted phenotype of lin28Tg CD8+ T cells by decreasing Tim-3- and increasing CD62L expression. Also, the ROS scavenger NAC reduces features of exhaustion but surprisingly doesn't increase CD62L (Figure 5c). Were different doses of NAC tested? The authors should compare the viability of untreated vs. NAC-treated CD8+ T cells to test whether NAC is toxic to lin28Tg CD8+ T cells.
- The authors treated P14+ CD8+ T cells from WT and lin28Tg mice with rapamycin and observed a reduction in ROS (5g). Why do lin28Tg CD8+ T cells not experience greater ROS reduction compared to WT cells when lin28Tg CD8+ T cells exhibit higher ROS levels in Figure 5b? Would this be clearer if not fold change but absolute values are analysed? Furthermore, inhibitor treatment only slightly reduces ROS – WT T cells have much lower ROS compared to lin28Tg CD8+ T cells treated with rapamycin. Did the authors also consider NAC treatment to minimize cellular ROS content, which would be more intuitive in light of NAC's role as a ROS scavenger?
- The authors attributed the TORC2 complex (RICTOR^{-/-}) but not the TORC1 complex (RAPTOR^{-/-}) of lin28Tg CD8+ T cells responsible for the induction of this terminally differentiated phenotype. This is interesting since rapamycin used before is expected to primarily act on mTORC1 (in an acute exposure). It would therefore be very interesting to confirm this finding by an experiment in which WT mice are injected with tumor cells (e.g. B16gp33) and adoptively transferred with P14+ CTLs derived from either i) lin28Tg mice, ii) lin28Tg RAPTOR^{-/-} mice or iii) lin28Tg RICTOR^{-/-} mice to compare tumor progression and overall survival.
- Did the authors also compare the tumor progression and survival of tumor-bearing mice adoptively transferred with no CTLs, lin28Tg CTLs, or lin28Tg CTLs treated with rapamycin and NAC to tumor-bearing mice receiving let-7Tg CTLs to make a direct comparison between lin28Tg rapamycin- and NAC-treated cells and let-7Tg CTLs (see Figure 5i)?
- The graphs in Figure 1b and Figure 5j compare survival of B16 tumor-bearing mice after adoptive transfer with either no CTLs or with lin28Tg CTLs. Whereas in Figure 1b, the lin28Tg CTLs offer an anti-tumor advantage to the recipient mice compared with the mice that did not receive CTLs, in Figure 5j, the survival curves of mice that received lin28Tg CTLs or no CTLs look almost identical. Could the authors please comment on these different results?
- The authors should please explain how the inhibited cytotoxic function of let-7Tg CTLs (Supplementary Figure 1b) fits to the increased anti-tumor capacity shown in Figure 1a-b.
- The authors show that lin28Tg CTLs exhibit higher ECAR compared with let-7Tg CTLs, suggesting that let-7 restricts glycolytic flux and thereby promotes memory formation

(Supplementary Figure 3d). It would be interesting to see if, conversely, OCR is suppressed in lin28Tg CTLs.

Minor points:

- The authors should discuss current relevant literature (e.g. Ando/Araki 2023 – JCI – DOI: <https://doi.org/10.1172/jci160025>).
- The authors should correct some subjective phrasing (e.g. “very low levels of ECAR”) (line 163).
- Lowercase letters (a-f) are used in their figures and corresponding figure legends, but refer to the figure components with uppercase letters (A-F).
- The authors do not cite or mention the gene signatures they used to perform GSEA in Figure 1f (Joshi/Kaech - Immunity - DOI: <https://doi.org/10.1016/j.immuni.2007.07.010>; Sarkar/Ahmed - J Exp Med. - DOI: <https://doi.org/10.1084/jem.20071641>).
- The authors describe effects of let-7 on PIP3 production (lines 283-285), which they do not show in Figure 5b.
- In Figure 1g, different fonts were used for the genes shown in the heatmap.
- The authors should standardize their nomenclature (e.g. “naive” vs “naïve”; “Rpl13a” vs. “RPI13A” (Figure 1i+j); genes analyzed via quantitative RT-PCR italicized (Figure 1i-j and Figure 2f) vs. normal letters (Figure 3b+d); “CD62L+CD44+” vs. “CD62L+CD44+”).
- In the graph shown in Figure 1a, the data points are represented by “border-colored dots”, but the legend shows “color-filled dots”.
- Figure 2e lacks a dotted line separating WT from GzmbCre+R26STOP-lin28-GFP mice as in Figure 2d.
- The authors show in Figure 3c that the population of SLECs, predominantly present in lin28Tg CD8+ T cells, decreases from day 9 to day 30, whereas MPECs, mainly detected in let-7Tg, accumulate in Rag2-/- host mice. It would be nice to plot these data over time to visualize differential survival.
- The authors claim that precursor exhausted T cells represent only a small population compared to terminally differentiated exhausted T cells (lines 78-80). This is not correct. The TPEX/TEX ratio is disease/context dependent.
- The authors do not clarify how let-7Tg mice are generated and how much let-7 they express.
- The authors speculate that the terminally differentiated/exhausted cells induced by lin28 are more prone to cell death (lines 208-210). Did they perform annexin staining that would support their hypothesis?
- How is tumor control achieved by let-7Tg CTLs despite suppressed metabolism (downregulation of Hk2 and glycolysis) and suppressed effector function (downregulation of Ifng and Gzmc) (see Figure 4c).
- The graphs shown in Supplementary Figure 1c are redundant to the graph shown in Figure 1a.
- The authors should indicate the specific time points at which the inhibitor injections occurred in the glycolytic flux assay shown in Supplementary Figure 3d.
- It is noticeable that the authors jump around between different Figures very frequently in the text of their manuscript, which interrupts the flow of reading.

Reviewer #3 (Remarks to the Author):

The authors present a sound study on the mechanistic role of let-7 on differentiation of murine cytotoxic T cells and the influence of let-7 expression on tumor growth.

They integrate in vitro and in vivo studies in subcutaneous tumor models with expression analyses on RNA and protein levels, thus forming a comprehensive picture of the let-7 mechanism and in vivo effect. Since let-7 miRNA are of current interest for cancer immunotherapy, I think this study will be of significance to the field. The methodology is sound and detailedly described.

The work mostly supports the conclusion drawn. However, there are some points, which I would like to highlight. Especially the connection between let-7 and its benefit for cancer immunotherapy is not addressed/discussed sufficiently, although it would greatly improve the importance of the study.

In general, I believe this study to be of interest for the readers of Nature Communications and would recommend a publication, if the following points are addressed:

1. Fig. 1a: Please show line graphs of averages and show in one graph. It would make the graph clearer and differences between individuals are already indicated by error bars and shown in suppl. material.

2. Fig. 1a and b: Are there significant differences in survival or tumor growth between the groups? Where statistical tests performed? If not, please do so and report in the figures.

3. Fig. 1d: Please show equal scales for x and y axes. Please also show a comparison as volcano plot between lin28 and let-7.

4. Line 191, concerning Fig. 2 d: The authors state that the anti-tumor response of lin28Tg CTLs is rescued by anti-PDL1 treatment "in a manner comparable of let7Tg TLS". I have several issues with that statement and experiment: 1. Why did the authors perform these experiments in a different tumor model (i.e. EL4 instead of B16 like before in Fig1)? 2. The anti-tumor response of the experiment shown in Fig. 2 and Fig. 1 cannot be compared with each other, especially not for different treatment groups, since two different tumor models were used. 3. Why is there no let.7 CTL group for the EL4 model? If the authors would like to show that lin28Tg CTLs are rescued by PDL1 treatment in order to receive an equal effect as the let7 group, they should include this group in their experiment. 4. Where statistical tests performed? If not, please do so and report in the figures.

5. Fig3D: The authors suggest that let-7 overexpression after the first 48h of stimulation had a smaller effect on expression of the relevant genes. Since there are no statistical analyses comparing the expression within the let-7 group over the different time point of overexpression (I assume the significant differences shown in the figure are for comparison between WT and Let7 group), one cannot make this assumption.

6. Fig. 5 I: Again, please show line graphs of averages and show in one graph. Which statistical test was performed (also for J)? Please indicate in the figure legend. Why is there no let-7 group to show that the lin28 Rapa+NAC group is rescued to a comparing anti-tumor response?

7. The authors nicely demonstrate in their manuscript that let-7 expression leads to an efficient anti-tumor response, development of a CTL memory phenotype and prevention of T

cell exhaustion. They indicate that this capability of let-7 expression in CTLs might be beneficial for immunotherapies, but they do not combine direct let-7 overexpression with checkpoint inhibition in a tumor model (only rescuing lin28-induced let-7 downregulation by antiPDL1 treatment). Could you include an experiment using let-7 overexpression in combination with checkpoint inhibition in an in vivo tumor model to make this final connection?

8. I think the discussion is too short. At least the role of let-7 in humans and especially in the context of immunotherapy, i.e. checkpoint inhibition, should be discussed:

There is one study in melanoma patients that received immunotherapy, which shows that expression of let-7 among other miRNAs leads to a decreased overall survival. To my expression, this contradicts the findings shown here and should at least be named in the discussion:

Huber V, Vallacchi V, Fleming V, Hu X, Cova A, Dugo M, Shahaj E, Sulsenti R, Vergani E, Filipazzi P, De Laurentiis A, Lalli L, Di Guardo L, Patuzzo R, Vergani B, Casiraghi E, Cossa M, Gualeni A, Bollati V, Arienti F, De Braud F, Mariani L, Villa A, Altevogt P, Umansky V, Rodolfo M, Rivoltini L. Tumor-derived microRNAs induce myeloid suppressor cells and predict immunotherapy resistance in melanoma. *J Clin Invest*. 2018 Dec 3;128(12):5505-5516. doi: 10.1172/JCI98060. Epub 2018 Nov 5. PMID: 30260323; PMCID: PMC6264733.

There are also other studies in humans showing that let-7 overexpression is beneficial for checkpoint inhibition. I think those studies should be included in the discussion to support the author's point. To name only one:

Yu, D., Liu, X., Han, G. et al. The let-7 family of microRNAs suppresses immune evasion in head and neck squamous cell carcinoma by promoting PD-L1 degradation. *Cell Commun Signal* 17, 173 (2019). <https://doi.org/10.1186/s12964-019-0490-8>

Kind regards
Kerstin Wennhold

REVIEWER COMMENTS

Reviewer #1 (Remarks to the Author):

The study by Wells et al describes an in-depth analysis of the role of let-7 miRNAs in mouse in vivo models. The authors performed a series of in-depth analysis of let-7 overexpressing and deficient CD8 T cells with cutting-edge technologies and mouse models. The authors discovered that let-7 deficiency, despite leading to a higher cytotoxic profile in vitro, results into subordinate tumor immunity while let-7 overexpression is superior. The in-depth studies showed that this is related to the induction of an exhaustion phenotype in case of let-7 overexpressing and a memory phenotype in case of let-7 overexpression, which relates the regulation of the mTOR pathway by let-7.

Some comments/suggestions remain.

We thank reviewers for many useful suggestions. Based on all reviewer's comments we generated additional data, substantially modified text (changes highlighted in red) and updated figures (main figures: 3 new subfigures were added and 7 subfigures were modified; supplementary subfigures: 8 new supplementary subfigures were added and one was modified). We believe that we addressed all the comments and incorporated almost all suggestions which improved the quality of our manuscript.

1. The experiment with anti-PDL1 (Fig 2D) is interesting but the tumor model shown in Figure 2d seems unresponsive to PD-1/PD-L1 blockade. Therefore it would be informative to know whether let-7 deficient CD8 T (lin28Tg) can also become less exhausted in a model that is sensitive to blockade of the PD-1/PD-L1 pathway (e.g. MC38), reflecting a clinical situation in which patients respond to PD-1/PD-L1 blockade. In addition, it would be of interest to know whether let-7 overexpressing CD8 T cells (let-7Tg) can also benefit from blocking the PD-1/PD-L1 pathway, in such models. These experiments are of interest for the potential of targeting let-7 miRNAs as a therapeutic tool (see also point 4).

We agree with the reviewer-1, MC38 is indeed responsive to PD-1/PDL1 checkpoint blockade. However, we did not want the endogenous response to mask the response of adoptively transferred T cells. For that reason, we chose to use EL-4 thymoma that is completely resistant to anti-PDL1 treatment. We have done experiments with EL-4 and let7Tg CTLs, but due to a very aggressive nature of this tumor, even let7Tg cells were not as effective as they were with B16. We exchanged the experiment with EL-4 for

another one that contained a group with let7Tg CTLs and put it in **new Supplementary figure 5f**. For the main figure we repeated the experiment using B16 tumor where we included two new additional groups: mice that received let7Tg CTLs with or without anti-PDL1. Lin28Tg CTLs responded to PDL1 treatment very similar to Let7Tg CTLs, while anti-PDL1 further improved response of Let7Tg CTLs (**please see the new figure-2e**).

2. The functionality of the Let-7 proficient and deficient CD8 T cells has not been addressed with respect to their polyfunctional cytokine profile (e.g. IFN-g, TNF, IL-2) at the protein level, which is considered as a hallmark of T cell (dys)functionality, especially in the context of exhaustion. The higher gene expression of IL-2 in lin28Tg (Fig 4d) seems unexpected in this respect.

We thank the reviewer-1 for such an excellent suggestion. Previously, we demonstrated that the expression of let-7 in CD8 T cells suppresses production of TNFa/IFNg during differentiation in vitro, such that Lin28Tg (let-7-deficient) CTLs produce more TNFa/IFNg, while let-7Tg CTLs produce less (please see Well AC et al., eLife 2017). Now, to respond to reviewer's comments, we included a new experiment, where frequency of polyfunctional donor CD8 T cells from TILs was assessed after adoptive transfer of P14 CTLs into B16gp33-tumor bearing host mice (**new Fig-2d**). After re-stimulation with P+I, we found that only let-7Tg TILs are polyfunctional and retain the ability to produce both cytokines TNFa and INFg. Indeed, these data strengthen our argument that let-7 expression prevents exhaustion of CTLs in TME and that terminally differentiated Lin28Tg CTLs are "switched off" by the immunosuppressive tumor microenvironment.

3. The discussion is quite short and it would be of interest if the authors would add discussion on let-7 miRNAs as a therapeutic tool as suggested by the authors (lines 339,340).

We agree with the reviewer's comment. We extended our discussion and included this point.

Other comments

4. It would be informative to indicate that the authors use mouse in vivo models, and accordingly this should be indicated in abstract and title.

We agree with the reviewer-1 and are happy to introduce this point into our abstract, although we are not sure about the title.

5. Typo: ACKNOLEDGMENTS > Acknowledgements

Thank you, we corrected it.

Reviewer #2 (Remarks to the Author):

In this study, Wells et al. follow up on prior observations of Let-7 regulating CD8+ T cell differentiation and function in vitro now in vivo. They analyze the contribution of let-7 miRNAs by examining let-7 deficient (lin28Tg) or overexpressing (let-7Tg) CTLs. Let-7 is known to be highly expressed by naïve CD8+ T cells to ensure quiescence and survival and is downregulated during transition into functional effector CD8+ T cells. The authors tested the anti-tumoral capacities of CTLs with different levels of let-7 expression injected into tumor-bearing mice and observed superior tumor control and survival of mice receiving let-7Tg CTLs. RNA-seq analysis of lin28Tg- and let-7Tg CTLs revealed divergent transcriptome profiles, with lin28Tg CTLs showing enhanced expression of effector/exhaustion-associated genes, while let-7Tg CTLs exhibited a memory-like program. In addition, lin28Tg CD8+ T cells exhibited increased mTOR signaling and ROS, which the authors followed up by performing FACS analysis of phosphorylated ERK, AKT, S6 proteins and ROS production. FACS analyses confirmed these findings of increased pERK, pAKT, pS6 and ROS content in lin28Tg CD8+ T cells compared to let-7Tg CD8+ T cells.

This study highlights the link between let-7 and the regulation of the quiescent naïve/memory program of CD8+ T cells and adds an investigation of a potential mechanism via suppression of the mTOR pathway. However, the authors do not show the molecular mechanism of how let-7 directly promotes the memory program and inhibits the effector program and how it regulates individual components of the mTOR signaling pathway.

The main criticism to the work is therefore that a lot of the effects of let-7 connected to the naïve/quiescent/memory T cell program are known. The conceptual advance here would be the mechanistic investigation regarding the metabolic signaling. However it is well known that naïve, memory and effector and exhausted T cell have different metabolic activity. The issue is that while let-7 can have multiple regulated targets in the metabolic signaling pathways, and this could translate into lower protein expression (not shown for most markers) if this would directly result in lower pathway activity is unclear. It is well possible from the data presented that let-7 could primarily act on regulating major differentiation cues (ie by regulating transcription factors) instead of directly modulating metabolic pathways. This question of direct let-7 involvement would be better addressed in an inducible let-7 system on different differentiation subsets.

We thank reviewers for many useful suggestions. Based on all reviewer's comments we generated additional data, substantially modified text (changes highlighted in red) and updated figures (main figures: 3 new subfigures were added and 7 subfigures were modified; supplementary subfigures: 8 new supplementary subfigures were added and one was modified). We believe that we addressed all the comments and incorporated almost all suggestions which improved the quality of our manuscript.

We thank the reviewer-2 for a very thorough review of our manuscript and constructive criticism. We believe that we dramatically improved the quality of the study by addressing most of the comments and suggestions. We agree with the reviewer-2 that the main limitation of our current study is the lack of the specific single target for let-7 miRNA in CD8 T cells that can "rescue" the observed phenotypes. Based on our analysis, none of the mRNAs of known transcription factors that play an important role in T cell differentiation (with the exception of Eomes, which we previously reported in our eLife 2017 paper) are predicted let-7 targets. In Fig-5, we provided the list of all predicted let-7 targets (based on RNAseq data) that are dysregulated in a let-7-dependent manner in CTLs. Since it is likely that many targets contribute to the observed phenotypes, we think that the identification of such a complex network is outside of the scope of our current manuscript but will be definitely addressed in our investigations in the future. Of note, we investigated the role of Eomes by generating Lin28Tg/Eomes-deficient CTLs, but the obtained data suggested no "rescue" function in vivo and will be included in the follow-up study, where we will focus on other direct let-7 targets as well (please see the graph).

Additional Points:

- The authors performed FACS staining and qPCR to validate the different programs of lin28Tg- and let-7Tg CTLs but provide only exemplary histograms for Tim-3, PD-1, CD39, CD160, 2B4, CD38, CD62L and CD127. In this regard, they should expand the data set of this part to draw conclusions based on cellular protein expression. In addition, other markers such as CD57, T-bet, Eomes, KLRG1, Blimp1, CD28, CD27, Foxo1, Bcl-6, CXCR3, STAT3 and STAT4 would also be informative for distinguishing between terminal differentiation vs. memory phenotype.

Great point! Following reviewer-2's suggestion, we stained CTLs for CD28, CD27, Foxo1 and CXCR3. In addition, we also analyzed the expression of TCF-1. Not all of these markers showed significant differences. We added the data to Supplementary figure 3c and included CXCR3 to Figure 1h.

In general, for the validation, we chose molecules that were differentially expressed according to our RNAseq data. Among the markers suggested by the reviewer, Bcl-6, Stat3 and Stat4 show no difference. T-bet, Eomes and Blimp1 were analyzed in our previous publication (Wells et al, eLife 2017). Unfortunately, KLRG1 can be detected after activation in vivo, but not in our in vitro experimental system, while CD57 is a marker for human cells.

- The authors note that lin28Tg CTLs exhibit a “terminally differentiated” phenotype. However, they respond to PD-1 checkpoint blockade (see Figure 2d), which seems contradictory since terminally differentiated CD8+ T cells are described as refractory to PD-1 treatment. Is the difference (lin28Tg vs. lin28Tg + aPDL1) in tumor volume statistically significant? Could the authors also provide the data for mice with EL-4gp33 tumors that received WT- or let-7Tg CTLs?

We agree that the term “terminally differentiated” is not perfect. For that reason, we are not claiming that these cells are terminally differentiated but rather have a “terminally differentiated phenotype” based on the markers. We are happy to add a short explanation in the text when we first use this term in the manuscript.

EL-4 is a very aggressive tumor that is completely resistant to PD-1/PDL1 checkpoint blockade and even let7Tg CTLs were not as effective as they were with B16. We exchanged the experiment with EL-4 for another one that contained a group with let7Tg CTLs and put it in new Supplementary figure 5f. The difference between Lin28Tg and Lin28Tg+aPDL1 in EL-4 model is statistically significant. For the main figure we repeated the experiment using B16 tumor where we included two additional groups: let7Tg CTLs with or without aPDL1, where the difference between Lin28Tg and Lin28Tg+aPDL1 is statistically significant (please see a new Figure 2e).

- Can the authors please explain why they observe differences in Id2, Cd244, Tcf7, Sell and Ccr7 expression in naïve CD8+ T cells and differences in Havcr2, Pdcd1, Tcf7, Sell and Ccr7 expression in untreated P14+ CTLs isolated from WT vs. iCre+R26STOP-lin28-GFP mice (Figure 2f)? Because these mice already exhibit basal differential gene expression, they may not be a suitable model to study the time kinetics of lin28 induction.

We agree that iCre+R26STOP-lin28-GFP mice may not be ideal, however this is the only model (which we made for this study) available to acutely deplete let-7 *in vivo*. The differences in marker expression in naïve and untreated cells could be explained by some differences in mouse genotypes, for example WT mice do not express GFP or iCre+R26STOP-lin28-GFP mice (P14+M2rtTA+tetO-Cre+R26^{Isl-lin28-GFP}+Rag2^{-/-}) may have some mixed background. We wanted to be fair and present the unnormalized data instead of a fold change because despite the basal differences in marker expression, the trend that demonstrates upregulation of markers of terminal differentiation and downregulation of memory phenotypic markers upon acute deletion of let7 in CD8 T cells, is still the same as in all other experiments shown in the paper. Therefore, we hope reviewer-2 will share our enthusiasm that these data confirm the main findings and should be presented as is.

- The authors demonstrate in Figure 3b that lin28Tg CD8+ T cells are more susceptible to IL-2 treatment, which triggers subsequent upregulation of exhaustion markers, whereas let-7Tg CD8+ T cells are more responsive to IL-15 intervention, which in turn leads to increased expression of memory markers. Did the authors also have the opportunity to examine the effect of IL-7 supplementation on let-7Tg- and lin28Tg CD8+ T cells, given that IL-7, along with IL-15, represents a hallmark cytokine in the formation and maintenance of memory CD8+ T cells?

We thank the reviewer-2 for this suggestion. We differentiated CTLs in the presence of IL-7 and obtained a similar result as with IL-15. Specifically, IL-7 was not able to reverse the phenotype of let-7-deficient cells. We added the data to a **new Supplementary figure 7a**.

- It is not intuitive to the reader of the manuscript why the authors frequently change markers indicating terminally differentiated vs. memory CD8+ T cells (e.g. Tim-3/PD-1, KLRG1/CD127, Tim-3/CD62L axes).

We thank the reviewer-2 for this comment, and we apologize for the confusion. Tim3/PD-1 staining in Figure 2a is a typical way of demonstrating terminally differentiated phenotype in TILs, while KLRG1/CD127 staining in Figure 3c is typical for discriminating between SLECs and MPECs. The reason we switched to Tim3 and CD62L in Figure 5 is because we wanted to demonstrate the features of both “terminal differentiation” and “memory” in CTLs simultaneously on one plot. **We added an explanation in the text.**

- The authors often make statements about T-cell function being affected by knockdown or overexpression of let-7, but only provide the gene expression of effector genes. It

would be more convincing if the authors could measure cytokine production (e.g. IFN, TNF, IL-2) or secretion of granzymes and perforins of stimulated let-7Tg- and lin28Tg CD8+ T cells by FACS analysis.

We thank the reviewer-2 for such an excellent suggestion. Previously, we demonstrated that the expression of let-7 in CD8 T cells suppresses production of TNF α /IFN γ during differentiation in vitro, such that Lin28Tg (let-7-deficient) CTLs produce more TNF α /IFN γ , perforin and GrA/B, while let-7Tg CTLs produce less (please see Wells AC et al., eLife 2017). Now, to respond to reviewer's comments, we include a new experiment, where frequency of polyfunctional donor CD8 T cells from TILs was assessed after adoptive transfer of P14 CTLs into B16gp33-tumor bearing host mice (new Fig-2d). After re-stimulation with P+I, we found that only let-7Tg TILs are polyfunctional and retain the ability to produce both cytokines TNF α and IFN γ . Indeed, these data strengthen our argument that let-7 expression prevents exhaustion of CTLs in the TME and that terminally differentiated Lin28Tg CTLs are "switched off" by the immunosuppressive tumor microenvironment.

- By RNA-seq analyses of naïve, 12h-activated, and 5-day-differentiated CD8+ T cells from lin28 or let-7Tg mice, the authors demonstrated a lin28Tg-driven induction of mTOR signaling and the ROS pathway (Figure 5a). To confirm these results, they performed FACS analyses and found increased levels of pERK, pAKT, pS6 and ROS in lin28Tg CD8+ T cells. To underline these interesting findings, the authors should measure MEK and PI3K with respect to mTOR signaling. Because lin28Tg CTLs (vs. let-7Tg CTLs) also showed upregulation of oxidative phosphorylation (Supplementary Figure 8a), the authors should additionally analyze mitochondrial bioenergetics, e.g. by staining MTG, MTDR, GSH, JC-1 or TMRE.

We agree with the reviewer-2. To demonstrate the increased activity of PI3K with respect to mTOR, we stained for Akt-Thr308 and observed more phosphorylation in activated Lin28Tg T cells. We added the data to a new Figure 5b. We also analyzed mitochondria of CTLs by staining with JC-1 and showed that Lin28Tg CTLs have more active mitochondria which is in line with overall increased metabolic state of let-7-deficient cells. The data is in a new Supplementary figure 3g.

- Subsequently, the authors show that mTOR inhibition by rapamycin reverses the exhausted phenotype of lin28Tg CD8+ T cells by decreasing Tim-3- and increasing CD62L expression. Also, the ROS scavenger NAC reduces features of exhaustion but surprisingly doesn't increase CD62L (Figure 5c). Were different doses of NAC tested?

The authors should compare the viability of untreated vs. NAC-treated CD8+ T cells to test whether NAC is toxic to lin28Tg CD8+ T cells.

We agree with the reviewer-2. We were also surprised that NAC activity is specific towards Tim3 but not CD62L, which most likely means that ROS is regulating Tim3 expression but is not upstream of CD62L. We tested different doses of NAC, and it was toxic to T cells (regardless of the origin) at 15mM. At 10mM it did not affect viability of T cells since recovery was always similar to DMSO treated controls. Specifically, NAC was not toxic to Lin28Tg T cells (please see the graph).

- The authors treated P14+ CD8+ T cells from WT and lin28Tg mice with rapamycin and observed a reduction in ROS (5g). Why do lin28Tg CD8+ T cells not experience greater ROS reduction compared to WT cells when lin28Tg CD8+ T cells exhibit higher ROS levels in Figure 5b? Would this be clearer if not fold change but absolute Values are analysed? Furthermore, inhibitor treatment only slightly reduces ROS – WT T cells have much lower ROS compared to lin28Tg CD8+ T cells treated with rapamycin. Did the authors also consider NAC treatment to minimize cellular ROS content, which would be more intuitive in light of NAC's role as a ROS scavenger?

We thank reviewer-2 for the comment and we are sorry for the confusion. The interplay between ROS and mTOR pathway is complex as mTOR is not the only activator of ROS and ROS themselves are also known to activate mTOR. We think that other factors are most likely contributing to the increased ROS in Lin28Tg T cells. With regards to the actual values, they differ from experiment to experiment, therefore we had to normalize the data so we could perform statistical analysis on pooled experiments. We showed that mTOR inhibition is critical at early time points after activation, while ROS inhibition is important later in differentiation. Therefore, the purpose of this experiment was not to just analyze ROS levels after NAC treatment, but to demonstrate that mTOR pathway is upstream of ROS. We believe that using rapamycin and then measuring ROS answers the question.

- The authors attributed the TORC2 complex (RICTOR^{-/-}) but not the TORC1 complex (RAPTOR^{-/-}) of lin28Tg CD8+ T cells responsible for the induction of this terminally differentiated phenotype. This is interesting since rapamycin used before is expected to primarily act on mTORC1 (in an acute exposure). It would therefore be very interesting to confirm this finding by an experiment in which WT mice are injected with tumor cells (e.g. B16gp33) and adoptively transferred with P14+ CTLs derived from either i) lin28Tg

mice, ii) lin28Tg RAPTOR^{-/-} mice or iii) lin28Tg RICTOR^{-/-} mice to compare tumor progression and overall survival.

Excellent point, we completely agree with the reviewer-2 (we really wanted to do it)! However, we decided not to perform such experiment because of three reasons:

1. Much milder phenotype of RAPTOR^{-/-} and RICTOR^{-/-} Lin28Tg CTLs in comparison to rapamycin/NAC-treated Lin28Tg CTLs (Fig-5).
2. Therapeutic relevance of rapamycin treatment.
3. Based on the previous publication (Ando/Araki 2023 – JCI – DOI: <https://doi.org/10.1172/jci160025> - suggested by the reviewer), mTOR inhibition during the expansion phase is very beneficial for the effector immune response, while prolonged/late mTOR suppression compromises it. This is why brief inhibition by rapamycin during activation/proliferation stage is effective in rescuing (partially) cytotoxic function of Lin28Tg CTLs in vivo.

- Did the authors also compare the tumor progression and survival of tumor-bearing mice adoptively transferred with no CTLs, lin28Tg CTLs, or lin28Tg CTLs treated with rapamycin and NAC to tumor-bearing mice receiving let-7Tg CTLs to make a direct comparison between lin28Tg rapamycin- and NAC-treated cells and let-7Tg CTLs (see Figure 5i)?

We agree with the reviewer-2, we repeated the experiment and added a new group that received let-7Tg CTLs (please see a **new Figure 5ij**).

- The graphs in Figure 1b and Figure 5j compare survival of B16 tumor-bearing mice after adoptive transfer with either no CTLs or with lin28Tg CTLs. Whereas in Figure 1b, the lin28Tg CTLs offer an anti-tumor advantage to the recipient mice compared with the mice that did not receive CTLs, in Figure 5j, the survival curves of mice that received lin28Tg CTLs or no CTLs look almost identical. Could the authors please comment on these different results?

Thank you for noticing! The reviewer-2 is absolutely right, we should be observing a slight inhibition of tumor growth in the presence of Lin28Tg CTLs. In fact, looking back, it was the only time when Lin28Tg CTLs exhibited no response at all. We repeated this experiment and replaced this figure and added the group that received Let-7Tg CTLs (please see the **new Figure 5ij**).

- The authors should please explain how the inhibited cytotoxic function of let-7Tg CTLs

(Supplementary Figure 1b) fits to the increased anti-tumor capacity shown in Figure 1a-b.

We thank the reviewer-2 for the comment and **modified the text for clarity**. Specifically, please see the first part of the results section and a new discussion part of the manuscript. Briefly, cytotoxic function of let-7Tg CTLs is indeed inhibited but not absent. We think that Let-7Tg CTLs during differentiation acquired properties of memory cells but not the effector cells. The important point that Let-7Tg CTLs do not express inhibitory receptors, which was proven to be detrimental to the immune responses in pathological conditions such as tumors and chronic infections, where CTLs with inhibitory receptors (like Lin28Tg CTLs) will be “switched off” and become exhausted. Therefore, once Let-7Tg CTLs (without Tim3, 2B4, CD160, low PD-1 etc.) are injected into tumor bearing mice, these cells are “invisible” to immunosuppressive tumor microenvironment and cannot be “switched off”. Please also note that Let-7Tg CTLs also have a superior survival (please see Figure 2bc, Figure 3a, Figure 5a and Supplementary Figure 3d).

- The authors show that lin28Tg CTLs exhibit higher ECAR compared with let-7Tg CTLs, suggesting that let-7 restricts glycolytic flux and thereby promotes memory formation (Supplementary Figure 3d). It would be interesting to see if, conversely, OCR is suppressed in lin28Tg CTLs.

Following reviewer-2 suggestion, we performed MitoStress test. We found that Lin28Tg CTLs have high OCR levels, while Let7Tg CTLs have low OCR. In fact, it goes in line with overall high metabolic status of lin28Tg CTLs. Moreover, these findings were also supported by pathway analysis of our RNAseq data (please see Supplementary Figure 8) We included these new results into a **new Supplementary Figure 3f**.

Minor points:

- The authors should discuss current relevant literature (e.g. Ando/Araki 2023 – JCI – DOI: <https://doi.org/10.1172/jci160025>).

We agree, it is a great paper, we included it in the discussion section to strengthen our arguments (please note, this paper came out after we submitted our manuscript in December of 2022).

- The authors should correct some subjective phrasing (e.g. “very low levels of ECAR”) (line 163).

We agree with the reviewer-2 and corrected the text.

- Lowercase letters (a-f) are used in their figures and corresponding figure legends, but refer to the figure components with uppercase letters (A-F).

We agree with the reviewer-2 and corrected the text.

- The authors do not cite or mention the gene signatures they used to perform GSEA in Figure 1f

(Joshi/Kaech - Immunity - DOI: <https://doi.org/10.1016/j.immuni.2007.07.010>

Sarkar/Ahmed - J Exp Med. - DOI: <https://doi.org/10.1084/jem.20071641>).

We absolutely agree with the reviewer-2 and included these important citations.

- The authors describe effects of let-7 on PIP3 production (lines 283-285), which they do not show in Figure 5b.

We agree with the reviewer-2 and corrected the text.

- In Figure 1g, different fonts were used for the genes shown in the heatmap.

We agree with the reviewer-2 and corrected fonts in figure-1.

- The authors should standardize their nomenclature (e.g. "naive" vs "naïve"; "Rp13a" vs. "RPI13A" (Figure 1i+j); genes analyzed via quantitative RT-PCR italicized (Figure 1i-j and Figure 2f) vs. normal letters (Figure 3b+d); "CD62L+CD44+" vs. "CD62L+CD44+").

We agree with the reviewer-2 and are happy to address comments above.

- In the graph shown in Figure 1a, the data points are represented by "border-colored dots", but the legend shows "color-filled dots".

We agree with the reviewer-2 and corrected the figure.

- Figure 2e lacks a dotted line separating WT from GzmbCre+R26STOP-lin28-GFP mice as in Figure 2d.

We agree with the reviewer-2 and corrected the figure.

- The authors show in Figure 3c that the population of SLECs, predominantly present in lin28Tg CD8+ T cells, decreases from day 9 to day 30, whereas MPECs, mainly detected in let-7Tg, accumulate in Rag2^{-/-} host mice. It would be nice to plot these data over time to visualize differential survival.

We generated a new supplementary figure 7b to show plotted data as requested.

- The authors claim that precursor exhausted T cells represent only a small population compared to terminally differentiated exhausted T cells (lines 78-80). This is not correct. The TPEX/TEX ratio is disease/context dependent.

We agree with the reviewer-2 that T_{pex}/T_{ex} ratio can be different and are happy to correct our statement in the text.

- The authors do not clarify how let-7Tg mice are generated and how much let-7 they express.

We thank the reviewer-2 for this comment and we are sorry for the confusion. These mice have been published in many studies, including from our group (Nat. Imm 2015, eLife 2017 and Front. Imm 2020). They are also available at JAX. Please note, we characterized the expression of let-7 transgene in naive, activated CD8 T cells and CTLs in our eLife 2017 paper which is cited in our manuscript.

- The authors speculate that the terminally differentiated/exhausted cells induced by lin28 are more prone to cell death (lines 208-210). Did they perform annexin staining that would support their hypothesis?

We thank reviewer-2 for this comment. Unfortunately, rapid clearance of apoptotic cells *in vivo* makes such simple experiment quite difficult and therefore less informative (not mentioning the effect of enzymatic digestion of tumors during TILs isolation). We believe that we performed a more informative experiment (Figure 2bc), where we rescued survival of Lin28Tg CTLs in TILs *in vivo* after transducing them with the pro-survival factor BCL-XL. We also observed rescue of Lin28Tg CTLs numbers with BCL-2-RV as well (not shown). In addition, we performed a cytokine withdrawal experiment, where we demonstrated a higher rate of cell death of Lin28Tg CTLs without IL-2 in comparison to Let-7Tg CTLs, which exhibited remarkable survival ability (Figure 3A). Finally, the pathway analysis of our RNAseq data also picked up apoptosis, as a hallmark pathway

that is upregulated in Lin28Tg cells (Figure 5a). Of note, we found similar survival defect in naïve T cells and T-helper cells (Front. Immunol 2019, 2020).

- How is tumor control achieved by let-7Tg CTLs despite suppressed metabolism (downregulation of Hk2 and glycolysis) and suppressed effector function (downregulation of Ifng and Gzmc) (see Figure 4c).

We thank the reviewer-2 for the comment and modified the text to increase the clarity. Specifically, please see the first part of the results section and a new discussion part of the manuscript. Briefly, we think that Let-7Tg CTLs during differentiation acquired properties of memory cells but not effector cells. The important point that Let-7Tg CTLs do not express inhibitory receptors, which are proven to be detrimental to the immune responses in pathological conditions such as tumors and chronic infections, where CTLs with inhibitory receptors (like Lin28Tg CTLs) will be “switched off” and become exhausted. Therefore, once Let-7Tg CTLs (without Tim3, 2B4, CD160, low PD-1 etc.) are injected into tumor bearing mice, these cells are “invisible” to immunosuppressive tumor microenvironment and cannot be “switched off”. Please also note that Let-7Tg CTLs also have superior survival (Fig-2bc and 3a).

- The graphs shown in Supplementary Figure 1c are redundant to the graph shown in Figure 1a.

We thank the reviewer-2 for the comment. It is true that it is the same experiment which is presented differently in Fig-1A and in Supplementary Figure-1C. We changed the appearance of the graph on Fig-1A. Please note that Supplement Figure-1C shows growth trajectory of the individual tumors, which are missing in Figure-1A.

- The authors should indicate the specific time points at which the inhibitor injections occurred in the glycolytic flux assay shown in Supplementary Figure 3d.

We thank the reviewer-2 for this comment and updated the figure.

- It is noticeable that the authors jump around between different Figures very frequently in the text of their manuscript, which interrupts the flow of reading.

We modified the text where it was possible and believe that the quality of the manuscript really improved after our revision and incorporation of reviewer’s suggestions.

Reviewer #3 (Remarks to the Author):

The authors present a sound study on the mechanistic role of let-7 on differentiation of murine cytotoxic T cells and the influence of let-7 expression on tumor growth.

They integrate in vitro and in vivo studies in subcutaneous tumor models with expression analyses on RNA and protein levels, thus forming a comprehensive picture of the let-7 mechanism and in vivo effect. Since let-7 miRNA are of current interest for cancer immunotherapy, I think this study will be of significance to the field. The methodology is sound and detailedly described.

The work mostly supports the conclusion drawn. However, there are some points, which I would like to highlight. Especially the connection between let-7 and its benefit for cancer immunotherapy is not addressed/discussed sufficiently, although it would greatly improve the importance of the study.

In general, I believe this study to be of interest for the readers of Nature Communications and would recommend a publication, if the following points are addressed:

We thank reviewers for many useful suggestions. Based on all reviewer's comments we generated additional data, substantially modified text (changes highlighted in red) and updated figures (main figures: 3 new subfigures were added and 7 subfigures were modified; supplementary subfigures: 8 new supplementary subfigures were added and one was modified). We believe that we addressed all the comments and incorporated almost all suggestions which improved the quality of our manuscript.

1. Fig. 1a: Please show line graphs of averages and show in one graph. It would make the graph clearer and differences between individuals are already indicated by error bars and shown in suppl. material.

According to the reviewer-3 suggestion, we changed the appearance of the graph in Fig. 1a.

2. Fig. 1a and b: Are there significant differences in survival or tumor growth between the groups? Where statistical tests performed? If not, please do so and report in the figures.

We thank reviewer-3 for noticing it, we updated the figures and added all requested statistical tests.

3. Fig. 1d: Please show equal scales for x and y axes. Please also show a comparison as volcano plot between lin28 and let-7.

We agree with the reviewer-3. We changed the scale for the volcano plots in a new Figure 1d and added a new Lin28/let7 volcano plot.

4. Line 191, concerning Fig. 2 d: The authors state that the anti-tumor response of lin28Tg CTLs is rescued by anti-PDL1 treatment “in a manner comparable of let7Tg TLS”. I have several issues with that statement and experiment: 1. Why did the authors perform these experiments in a different tumor model (i.e. EL4 instead of B16 like before in Fig1)? 2. The anti-tumor response of the experiment shown in Fig. 2 and Fig. 1 cannot be compared with each other, especially not for different treatment groups, since two different tumor models were used. 3. Why is there no let.7 CTL group for the EL4 model? If the authors would like to show that lin28Tg CTLs are rescued by PDL1 treatment in order to receive an equal effect as the let7 group, they should include this group in their experiment. 4. Where statistical tests performed? If not, please do so and report in the figures.

We understand reviewer-3 concerns. We are happy to correct our experiment and explain our reasoning. We chose to use EL-4 thymoma that is completely resistant to anti-PDL1 treatment because we did not want the endogenous response to mask the response of adoptively transferred T cells (even B16 has a small response to checkpoint blockade with anti-PD-L1). We have done experiments with EL-4 and let7Tg CTLs, but due to a very aggressive nature of this tumor, even let7Tg CTLs were not as effective as they are against B16. Per the reviewer’s request, we replaced the experiment with EL-4 for another one that contains a group with let7Tg CTLs and put it in Supplementary figure 5f. For the main figure we did one more experiment using B16 tumor where we included two additional groups: let7Tg CTLs with or without anti-PDL1 (please see a new Figure 2e). Lin28Tg CTLs responded to anti-PDL1 treatment very similar to Let7Tg CTLs. Please note, that anti-PDL1 treatment further improved the response of Let7Tg CTLs. We also modified text accordingly.

5. Fig3D: The authors suggest that let-7 overexpression after the first 48h of stimulation had a smaller effect on expression of the relevant genes. Since there are no statistical analyses comparing the expression within the let-7 group over the different time point of overexpression (I assume the significant differences shown in the figure are for comparison between WT and Let7 group), one cannot make this assumption.

We agree with the reviewer-3 that we did not formally compare early versus late induction of let-7 expression in let-7Tg CD8 T cells during differentiation in vitro. We revised our figure and made such a comparison (please see a new Supplementary Figure 7d)

6. Fig. 5 I: Again, please show line graphs of averages and show in one graph. Which statistical test was performed (also for J)? Please indicate in the figure legend. Why is there no let-7 group to show that the lin28 Rapa+NAC group is rescued to a comparing anti-tumor response?

Following the reviewer-3 suggestion, we changed the appearance of the graph in Figure 5i. We performed two-way ANOVA with Sidak's multiple comparison test on tumor growth curves and log-rank Mantel-Cox test on survival curves and added the information to the figures. We repeated the experiment with rapamycin and NAC and included the let-7 group. Rapamycin and NAC significantly improved the anti-tumor response of Lin28Tg T cells but did not rescue it to the level of let-7Tg cells.

7. The authors nicely demonstrate in their manuscript that let-7 expression leads to an efficient anti-tumor response, development of a CTL memory phenotype and prevention of T cell exhaustion. They indicate that this capability of let-7 expression in CTLs might be beneficial for immunotherapies, but they do not combine direct let-7 overexpression with checkpoint inhibition in a tumor model (only rescuing lin28-induced let-7 downregulation by antiPDL1 treatment). Could you include an experiment using let-7 overexpression in combination with checkpoint inhibition in an in vivo tumor model to make this final connection?

We thank reviewer-3 for this suggestion. Indeed, it is a great point that can strengthen our paper. We added two groups to the experiment shown in a new Figure 2e: let7Tg CTLs with or without anti-PDL1. Please note, anti-PDL1 treatment further improved response of Let7Tg CTLs making our story even more exciting!

8. I think the discussion is too short. At least the role of let-7 in humans and especially in the context of immunotherapy, i.e. checkpoint inhibition, should be discussed:

There is one study in melanoma patients that received immunotherapy, which shows that expression of let-7 among other miRNAs leads to a decreased overall survival. To my expression, this contradicts the findings shown here and should at least be named in the discussion:

Huber V, Vallacchi V, Fleming V, Hu X, Cova A, Dugo M, Shahaj E, Sulsenti R, Vergani E, Filipazzi P, De Laurentiis A, Lalli L, Di Guardo L, Patuzzo R, Vergani B, Casiraghi E, Cossa M, Gualeni A, Bollati V, Arienti F, De Braud F, Mariani L, Villa A, Altevogt P, Umansky V, Rodolfo M, Rivoltini L. Tumor-derived microRNAs induce myeloid suppressor cells and predict immunotherapy resistance in melanoma. *J Clin Invest*. 2018 Dec 3;128(12):5505-5516. doi: 10.1172/JCI98060. Epub 2018 Nov 5. PMID: 30260323; PMCID: PMC6264733.

There are also other studies in humans showing that let-7 overexpression is beneficial for checkpoint inhibition. I think those studies should be included in the discussion to support the author's point. To name only one: Yu, D., Liu, X., Han, G. et al. The let-7 family of microRNAs suppresses immune evasion in head and neck squamous cell carcinoma by promoting PD-L1 degradation. *Cell Commun Signal* 17, 173 (2019). <https://doi.org/10.1186/s12964-019-0490-8>

We absolutely agree with the reviewer-3. We extended the discussion section and included suggested references.

REVIEWER COMMENTS

Reviewer #1 (Remarks to the Author):

The authors have addressed all comments by revising parts of the text, adding new experiments including functional assays and tumor challenges, and by extending the discussion.

Reviewer #2 (Remarks to the Author):

In the revised work the authors have addressed a number of concerns raised, which has improved the manuscript. However a key issue remains which is that while they show a link between let-7 and memory programming opposed to terminal differentiation they do not show mechanistically how let 7 acts on metabolism pathways.. Thus, the work still does not show the molecular mechanism of how let-7 directly promotes the memory program and inhibits the effector program and how it regulates individual components of the mTOR signaling pathway. So the work is essentially underlining the key findings from their 2017 paper. The molecular mechanisms are certainly difficult to tease apart particularly with regards of how broad let7 targets are and how metabolism is interconnected with differentiation cues but the authors also didn't perform conclusive experiments that might have addressed these points more clearly and now refer to a discussion of recent data by Ando/Araki et al JCI 2023. So overall, since the experiments using mtor inhibition and ROS scavenging do not clarify the issue the mechanistic statement in the title clearly is overstating and misleading. Rather a title such as "Let-7 enhances anti-tumor CD8 T cell responses by supporting the memory program associated with metabolic quiescence and antagonizing terminal differentiation." would be more accurate.

There remain other minor issues:

- a) Statistics please check if normal distribution can be assumed for the parametric testing performed. It might be more appropriate to consider employing nonparametric statistical tests for low n experiments.
- b) Typo/Nomenclature uniformity: Examples of inconsistencies include the use of "naïve" versus "naive," "PD-1" versus "PD1," "anti" versus "a," "Tim-3" versus "Tim3" versus "Tim 3," "Let7-Tg" versus "Let7Tg," the superscript "3" in "mm3," and "IFN γ " versus "IFNg."
- c) In the experiments now new in Supplementary Figure 3f-g, it is interesting that the basal oxygen consumption rate (OCR) of WT and lin28Tg mice appears remarkably similar, a notable contrast emerges in the JC-1 readout. In this regard, lin28Tg mice exhibit a significantly higher JC-1 signal in comparison to WT mice. A comparison with a different membrane potential probe (e.g., TMRE) might clarify why this discrepancy is found
- d) The information that cells were stimulated with PMA and ionomycin for a duration of 4 hours for intracellular cytokine staining should be added to figure 2.
- e) In their detailed response to our comment regarding the viability of NAC treated cells, the authors write that the NAC/DMSO ratio remains consistent across all cell lines. However, the graph illustrates variations in the mean values from WT (mean: approx. 1.5) to let-7Tg (mean: approx. 2.5), which looks distinct to readers. It would be beneficial if the authors could include a statistical analysis and add this data to the manuscript since the data indicate that the chosen NAC dose induces some limited cytotoxicity.

Reviewer #3 (Remarks to the Author):

The authors addressed the reviewers comments in their revision and performed extensive additional experiments. Therefore, I now find it of sufficient quality for Nature Communications and recommend publication.

SECOND REVISION

We are delighted to see that reviewer-1 and reviewer-3 agreed with all our changes and already suggested to accept the manuscript. We would like to thank the reviewer-2 for additional useful suggestions, which we fully implemented and even further improved our manuscript.

REVIEWER COMMENTS

Reviewer #2 (Remarks to the Author):

So overall, since the experiments using mtor inhibition and ROS scavenging do not clarify the issue the mechanistic statement in the title clearly is overstating and misleading. Rather a title such as "Let-7 enhances anti-tumor CD8 T cell responses by supporting the memory program associated with metabolic quiescence and antagonizing terminal differentiation." would be more accurate.

We liked the reviewer's idea, therefore we changed the title to "Let-7 enhances anti-tumor CD8 T cell responses by supporting the memory program associated with metabolic quiescence and antagonizing terminal differentiation".

1. Statistics please check if normal distribution can be assumed for the parametric testing performed. It might be more appropriate to consider employing nonparametric statistical tests for low n experiments.

We thank the reviewer for the suggestion. Parametric testing seems to be routinely used in similar applications (<https://www.nature.com/articles/s41590-020-0733-2>, <https://www.nature.com/articles/s41590-022-01369-x>). We have used Mann-Whitney tests to reanalyze data that may have been influenced by abnormal data distributions (namely, figures 2 a, b, and d), though it does not change our findings.

2. Typo/Nomenclature uniformity: Examples of inconsistencies include the use of "naïve" versus "naive," "PD-1" versus "PD1," "anti" versus "a," "Tim-3" versus "Tim3" versus "Tim 3," "Let7-Tg" versus "Let7Tg," the superscript "3" in "mm3," and "IFN γ " versus "IFNg."

Thank you for noticing, we revised the text and figures.

3. In the experiments now new in Supplementary Figure 3f-g, it is interesting that the basal oxygen consumption rate (OCR) of WT and lin28Tg mice appears remarkably similar, a notable contrast emerges in the JC-1 readout. In this regard, lin28Tg mice exhibit a significantly higher JC-1 signal in comparison to WT mice. A comparison with a different membrane potential probe (e.g., TMRE) might clarify why this discrepancy is found.

Per reviewer request, we repeated our experiment using TMRE probe and obtained similar results, although less dramatic as with JC-1. We included both experiments in the manuscript. Please see a new Supplementary figure-3h.

4. The information that cells were stimulated with PMA and ionomycin for a duration of 4 hours for intracellular cytokine staining should be added to figure 2.

We included the requested information.

5. In their detailed response to our comment regarding the viability of NAC treated cells, the authors write that the NAC/DMSO ratio remains consistent across all cell lines. However, the graph illustrates variations in the mean values from WT (mean: approx. 1.5) to let-7Tg (mean: approx. 2.5), which looks distinct to readers. It would be beneficial if the authors could include a statistical analysis and add this data to the manuscript since the data indicate that the chosen NAC dose induces some limited cytotoxicity.

We included requested information in a new Supplementary figure-9a.

REVIEWERS' COMMENTS

Reviewer #2 (Remarks to the Author):

The authors have clarified my remaining points. I have no further concerns.

Reviewer #3 (Remarks to the Author):

The authors addressed the reviewers concerns and I recommend the manuscript for publication.

THIRD REVISION

Since the reviewers have no more concerns left we would like just to thank them for many useful incites that made our manuscript better.